# The impact of individual perceptual and cognitive factors on collective states in a data-driven fish school model

**Weijia Wang**[1,2,3], **Ramón Escobedo**[2], **Stéphane Sanchez**[3], **Clément Sire**[4], **Zhangang Han**[1], **Guy Theraulaz**[2]*

**1** School of Systems Science, Beijing Normal University, Beijing, China, **2** Centre de Recherches sur la Cognition Animale, Centre de Biologie Intégrative (CBI), Centre National de la Recherche Scientifique (CNRS) & Université de Toulouse Paul Sabatier, Toulouse, France, **3** Institut de Recherche en Informatique de Toulouse (IRIT), Université de Toulouse Capitole, Toulouse, France, **4** Laboratoire de Physique Théorique, CNRS & Université de Toulouse Paul Sabatier, Toulouse, France

* guy.theraulaz@univ-tlse3.fr

**Data Availability Statement:** All relevant data are within the manuscript and its Supporting information files.

## Abstract

In moving animal groups, social interactions play a key role in the ability of individuals to achieve coordinated motion. However, a large number of environmental and cognitive factors are able to modulate the expression of these interactions and the characteristics of the collective movements that result from these interactions. Here, we use a data-driven fish school model to quantitatively investigate the impact of perceptual and cognitive factors on coordination and collective swimming patterns. The model describes the interactions involved in the coordination of burst-and-coast swimming in groups of *Hemigrammus rhodostomus*. We perform a comprehensive investigation of the respective impacts of two interactions strategies between fish based on the selection of the most or the two most influential neighbors, of the range and intensity of social interactions, of the intensity of individual random behavioral fluctuations, and of the group size, on the ability of groups of fish to coordinate their movements. We find that fish are able to coordinate their movements when they interact with their most or two most influential neighbors, provided that a minimal level of attraction between fish exist to maintain group cohesion. A minimal level of alignment is also required to allow the formation of schooling and milling. However, increasing the strength of social interactions does not necessarily enhance group cohesion and coordination. When attraction and alignment strengths are too high, or when the heading random fluctuations are too large, schooling and milling can no longer be maintained and the school switches to a swarming phase. Increasing the interaction range between fish has a similar impact on collective dynamics as increasing the strengths of attraction and alignment. Finally, we find that coordination and schooling occurs for a wider range of attraction and alignment strength in small group sizes.

**Funding:** GT, RE, SS and CS were supported by the Agence Nationale de la Recherche (ANR-20-CE45-0006-1). WW was funded by a grant from the China Scholarship Council (CSC N° 201906040126). ZH was supported by the National Natural Science Foundation of China (Grant No. 62176022). The funders had no role in study design, data collection and analysis, decision to publish, or preparation of the manuscript.

**Competing interests:** The authors have declared that no competing interests exist.

## Author summary

In fish schools, social interactions allow individuals to coordinate their movements and their modulation shape the emergent patterns of collective behavior. Here, we use a data-driven fish school model to investigate the impact of perceptual and cognitive factors on collective swimming patterns in the rummy-nose tetra (*H. rhodostomus*). In this species, fish only pay attention to one or two neighbors that exert the largest influence on their behavior and the interactions consist for a fish to be attracted and aligned with these neighbors. We show that there must exist a minimum level of alignment and attraction between fish to maintain group cohesion and allow the emergence of schooling and milling. Moreover, increasing the interaction range has a similar impact on collective dynamics as increasing the strength of social interactions. However, when the intensity of these interactions becomes too strong, fish can no longer coordinate their swimming and the school adopts a swarming behavior. Our results also show that a moderate level of behavioral fluctuations in fish can induce spontaneous transitions between schooling and milling. Finally, in this species that performs burst-and-coast swimming, we find that coordination occurs for a wider range of interaction strengths only in small group sizes.

## 1 Introduction

Many organisms living in groups or societies, from bacteria to vertebrates, are able to collectively coordinate their movements [1, 2]. These collective behaviors have important functional consequences for group members. For instance, coordinated movements of fish improve the foraging efficiency of individuals [3, 4], increase their survival under predation risk [3, 5], and in some cases allow them to save energy [5–10]. Analyzing the behavioral and cognitive mechanisms that govern these coordination phenomena in different species is a crucial step to understand the evolution of sociality and more generally the evolution of biological complexity [11–15].

For a long time, studies on the collective movements in animals groups mainly relied on the construction and analysis of mathematical models often without a direct link with experimental observations, due to the difficulty of acquiring precise data on these phenomena and translate them into faithful mathematical models [16–22]. Several phenomenological models based on specific rules can qualitatively reproduce the collective motion of real fish schools. However, the main problem remains the lack of experimental validation of the hypotheses these models are based on, and particularly those concerning the behavioral rules and interactions at the individual scale. In fact, very different rules can produce similar collective states.

In recent years, the development of new computerized methods based on learning algorithms has enabled the automated tracking of the behavior of animals moving in groups [23–28]. These techniques have been used in particular for the analysis of the collective movements of swarms of locusts and midges [29, 30], schools of fish in experimental tanks [31, 32], flocks of starlings [33, 34], flocks of sheep [35, 36], or human crowds [37, 38]. By improving the amount and accuracy of the data available on social interactions and their effects on individual behavior [39–42], these techniques have also paved the way for the development of models of collective movements that can be both quantitative and predictive [31, 43, 44].

One crucial step to develop such models requires knowing the social interactions that are involved in the coordination of movements. We have recently introduced a general method to extract from individuals' trajectories the social interaction functions between two individuals that are required to achieve coordinated motion [45]. Using large sets of tracking data, we

used this method to reconstruct and model the social interactions between individuals in rummy-nose tetra (*H. rhodostomus*) and zebra fish (*D. rerio*). These species perform a burst-and-coast type of swimming characterized by sequences of sudden increase in speed followed by a mostly passive gliding period. This particular type of swimming makes it possible to analyze individual trajectories as series of discrete behavioral decisions in time and space. These decisions correspond to significant variations in fish heading that occur exactly at the onset of the bursting phase. The effect of social interactions on a fish heading can then be precisely measured. In particular, one can quantify the strength of attraction, repulsion, and alignment behavior resulting from these interactions as a function of the distance between fish and their relative positions and orientations.

These analyses revealed that the range of the social interactions and the individuals' perception of their environment were quite different in these two species. In particular, we found that in *D. rerio*, not only the range of the attraction and alignment interactions was much shorter but also the perception was strongly reduced in front of a fish, both features contributing to a much weaker coordination of motion in pairs of fish in this species in comparison with *H. rhodostomus*. Certain characteristics of the natural environment in which fish swim, such as water turbidity [46–48] or light intensity [49], can also affect the perception of individuals and, consequently, the social interactions. Studies have shown that, as turbidity increases, fish form and maintain groups less efficiently, inter-fish distance increases, and groups are less able to coordinate their collective behavior [46–48]. All these results show that the sensory perception characteristics of individuals have a deep impact on their ability to coordinate their movements which in turn conditions the way information propagates within a group [50, 51] and the emergence of other groups level properties such as collective sensing [52, 53].

The limited attention capacity of individuals is another fundamental cognitive constraint on the coordination processes [54]. Several studies have shown that animals may focus their attention on a small subset of their neighbors, typically one or two, whose identity regularly changes with time [55–59]. The analysis of individual trajectories combined with the techniques used to reconstruct and model interactions between individuals also make it possible to measure at each moment the respective influence exerted by the neighbors of an individual on its movements within a group. Using this technique, Lei *et al.* have shown that in small groups of rummy-nose tetra, individuals typically interact with their two most influential neighbors, *i.e.*, the ones that have the strongest contribution to their heading variation [60]. Focusing the attention on those neighbors reduces significantly the amount of information that needs to be processed by a fish and avoids cognitive overload. However, the consequences of such information filtering at the individual level on collective states in fish species with a burst-and-coast swimming mode remains largely unknown.

Here, we study the impact of these individual perceptual and cognitive factors on fish coordination and on the collective phases that emerge at the group level from social interactions between fish. To do that, we use the data-driven computational model developed by Calovi *et al.* (2018) [44] and Lei *et al.* (2020) [60] that describes the interactions involved in the coordination of burst-and-coast swimming in groups of *H. rhodostomus*. With this model, we perform a comprehensive investigation of the respective impacts of two interactions strategies based on the selection of the most or the two most influential neighbors, of the range and intensity of social interactions, of the intensity of individual random fluctuations, and of the group size, on the ability of groups of 100 fish swimming in an unbounded space to coordinate their movements. We then construct the corresponding phase diagrams of the model that allows us to clearly identify the different phases resulting from interactions between fish, and investigate the impact of the parameters of the model on these collective states.

## 2 Materials and methods

### 2.1 Computational model

**2.1.1 Individual burst-and-coast swimming.** *H. rhodostomus* displays a "burst-and-coast" mode of swimming characterized by sequences of sudden accelerations called "kicks" followed by a quasi-passive deceleration period during which the fish glides along a near straight line until the next kick. Consecutive kicks of a single fish do not necessarily have the same length, the same duration, nor the same speed [61]. When swimming in groups, kicks of different fish are asynchronous and of different length, duration and speed.

These features are explicitly considered in our model, where we assume that individual fish makes the decision of changing direction precisely at the onset of each kick, and kicks characteristics are randomly sampled from distributions extracted directly from our experimental observations [44]. Note that the burst-and-coast swimming mode of *H. rhodostomus* was found to be directly responsible for the fact that the fish were most often swimming along the walls of the circular tanks considered in [44].

We denote by $t_i^n$ the instant of time at which the *n*-th kick of fish *i* starts, and by $\tau_i^n$ and $l_i^n$ the duration and the length of this kick, respectively. The position and velocity of fish *i* at this kicking time are $\vec{u}_i^n = (x_i^n, y_i^n)$ and $\vec{v}_i^n = v_i^n(\cos \phi_i^n, \sin \phi_i^n)$ respectively, where $v_i^n = \|\vec{v}_i^n\|$ is the speed at the beginning of the kick and $\phi_i^n$ is the angle that the velocity vector forms with the horizontal line, which remains unchanged all along the gliding phase of the kick. Positive angles are defined in the counter-clockwise direction, with respect to the positive semiaxis of abscissas of the global system of reference centered in $(0, 0)$. The angle $\phi_i^n$ thus determines fish heading, and heading changes are denoted by $\delta\phi_i^n$ [44]. See Fig 1A.

Fig 1B shows the *n*-th kick of fish *i*. The fish arrives at $\vec{u}_i^n$ at time $t_i^n$ with a velocity $\vec{v}_i^n$. Then, the fish collects the information of its instantaneous relative position and heading with

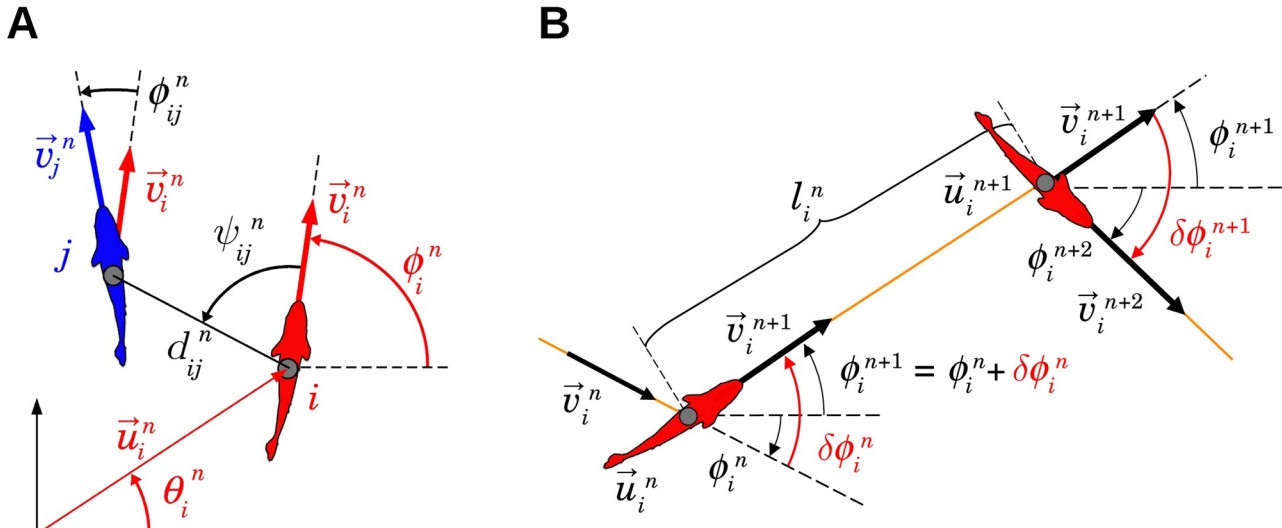

**Fig 1. State variables of fish and burst-and-coast swimming mode.** (A) Individual state variables (red) of fish *i* and social state variables (black) of fish *i* with respect to fish *j* at the instant $t_i^n$ when fish *i* performs its *n*-th kick: $\vec{u}_i^n$ and $\vec{v}_i^n$ are the position and velocity vectors of fish *i*, $\theta_i^n$ and $\phi_i^n$ are the angles that these respective vectors form with the horizontal line, $\vec{v}_j^n$ is the velocity vector of fish *j* at the instant of time of the *n*-th kick of fish *i*, $d_{ij}^n$ is the distance between fish *i* and fish *j*, $\psi_{ij}^n$ is the angle with which fish *j* is perceived by fish *i*, and $\phi_{ij}^n = \phi_j^n - \phi_i^n$ is the heading difference between both fish. (B) Description of the *n*-th kick performed by fish *i*, moving from $\vec{u}_i^n$ at time $t_i^n$ to $\vec{u}_i^{n+1}$ at time $t_i^{n+1}$. Kick length is denoted by $l_i^n$. Orange lines represent the fish trajectory, black wide arrows denote velocity vector, and curved arrows denote angles. The variation of the heading angle of fish *i* at time $t_i^n$ is $\delta\phi_i^n$. The fish heading during its *n*-th kick is $\phi_i^{n+1} = \phi_i^n + \delta\phi_i^n$. Red angles are the heading variation at the kicking instants $t_i^n$ and $t_i^{n+1}$.

respect to the other agents, and selects the length $l_i^n$ and duration $\tau_i^n$ of its $n$-th kick, and its change of direction, $\delta\phi_i^n$. During its $n$-th kick, the heading angle $\phi_i^{n+1} = \phi_i^n + \delta\phi_i^n$ remains unchanged until the onset of the next kick.

The state of fish $i$ at the end of its $n$-th kick, at time $t_i^{n+1}$, is given by

$$t_i^{n+1} = t_i^n + \tau_i^n, \tag{1}$$

$$\vec{u}_i^{\,n+1} = \vec{u}_i^{\,n} + l_i^n\, \vec{e}\left(\phi_i^{n+1}\right), \tag{2}$$

$$\phi_i^{n+1} = \phi_i^n + \delta\phi_i^n, \tag{3}$$

where $\vec{e}\left(\phi_i^{n+1}\right)$ is the unitary vector pointing in the heading direction $\phi_i^{n+1}$.

Kick lengths and durations $l_i^n$ and $\tau_i^n$ are sampled from bell-shaped distributions obtained in the experiments, where the mean values $l$ = 7 cm and $\tau$ = 0.5 s were found [44], as well as the speed maxima measured right after a kick, whose mean is $v_0$ = 14 cm/s. During the gliding phase of the kick, fish speed decreases quasi-exponentially with a relaxation time measured to be $\tau_0 \approx 0.8$ s, so that $v(t) = v_0 e^{-t/\tau_0}$ until the next kicking instant. All these features are implemented in the model [44]. Heading angle changes at the kicking instants result from the additive combination of the behavioral reaction induced by the social interaction with other fish, denoted by $\delta\phi_{S,i}^n$, and the spontaneous fluctuations in the motion of the fish, reproduced by a random Gaussian white noise $\delta\phi_{R,i}^n$:

$$\delta\phi_i^n = \delta\phi_{S,i}^n + \delta\phi_{R,i}^n. \tag{4}$$

**2.1.2 Social interactions between fish.** We assume that social influence is mediated by pairwise interactions, and that, in the case of several fish swimming together, the resulting effect is given by the sum of the pairwise interactions among a certain number of neighbors $k$. The function $\delta\phi_{ij}$ depends only on the relative state of fish $j$ with respect to the focal fish $i$. This state is determined by the triplet $(d_{ij}, \psi_{ij}, \phi_{ij})$, where $d_{ij}$ is the distance between fish, $\psi_{ij} = \theta_{ij} - \phi_i$ is the angle with which fish $i$ perceives fish $j$, where $\theta_{ij}$ is the angle that the vector going from $i$ to $j$ forms with the horizontal line, and $\phi_{ij} = \phi_j - \phi_i$ is the relative heading, which is a measure of the alignment between fish $i$ and $j$ (see Fig 1A). Thus:

$$\delta\phi_{S,i}^n = \sum_{j=1}^{k} \delta\phi_{ij}^n(d_{ij}^n, \psi_{ij}^n, \phi_{ij}^n). \tag{5}$$

The derivation of the shape and intensity of the functions $\delta\phi_{ij}$ and $\delta\phi_{R,i}$ is based on physical principles of symmetry of the angular functions and on a reconstruction procedure from trajectory data which is detailed in Calovi *et al.* (2018) [44] for the case of *H. rhodostomus* and in Escobedo *et al.* (2020) [45] for the general case of moving animal groups. In these works, we found that the effect of pairwise interactions $\delta\phi_{ij}$ consists of the sum of two contributions, an attraction force and an alignment force, each one described by the product of decoupled functions of the state variables:

$$\delta\phi_{ij}(d_{ij}, \psi_{ij}, \phi_{ij}) = \delta\phi_{Att}(d_{ij}, \psi_{ij}, \phi_{ij}) + \delta\phi_{Ali}(d_{ij}, \psi_{ij}, \phi_{ij}), \tag{6}$$

with

$$\delta\phi_{Att}\left(d_{ij}, \psi_{ij}, \phi_{ij}\right) = F_{Att}(d_{ij})\, O_{Att}(\psi_{ij})\, E_{Att}(\phi_{ij}), \tag{7}$$

$$\delta\phi_{Ali}(d_{ij}, \psi_{ij}, \phi_{ij}) = F_{Ali}(d_{ij})\, E_{Ali}(\psi_{ij})\, O_{Ali}(\phi_{ij}), \tag{8}$$

where *F* stands for the intensity of the interactions, and *O* and *E* for the odd or even symmetry in the angular dependence respectively.

We also found that the stochastic heading variation can be described by

$$\delta\phi_{\mathrm{R},i}^{n} = \gamma_{\mathrm{R}} g_{i}^{n}, \tag{9}$$

where $\gamma_{\mathrm{R}}$ is the noise intensity and $g_{i}^{n}$ is a random number sampled at time $t_{i}^{n}$ for this specific kick of the fish *i* from a standard normal distribution (zero mean, unit variance).

**2.1.3 Social interaction strategy.** In the model, the social interaction strategy determines how many neighbors and which ones are selected by the focal fish to interact with. Our recent results obtained with the model above presented have shown that only two neighbors, the *most influential* ones, are required to reproduce real fish behavioral patterns swimming in circular tanks [60].

The *influence* $\mathcal{I}_{ij}(t)$ that fish *j* exerts on fish *i* at time *t* is defined as the absolute value of the contribution of fish *j* to the instantaneous heading change of fish *i*:

$$\mathcal{I}_{ij}(t) = |\delta\phi_{ij}(t)|. \tag{10}$$

In a group of more than 2 fish, the *most influential neighbor* is the fish having the highest influence on the focal fish heading variation, *i.e.*, the highest value of $\mathcal{I}_{ij}$. The influence of fish *j* on fish *i* can change from one kick to another; the identity of the most influential neighbor of fish *i* changes accordingly.

This influence takes into account not only the distance to the neighbors, but also the anisotropy of the field of perception induced by the relative position of the influential fish and the degree of alignment between both fish.

For *H. rhodostomus*, the pairwise interaction functions are [60]:

$$F_{\mathrm{Att}}(d) = \gamma_{\mathrm{Att}} \frac{d/d_{\mathrm{Att}} - 1}{1 + (d/l_{\mathrm{Att}})^{2}}, \tag{11}$$

$$O_{\mathrm{Att}}(\psi) = 1.395\sin(\psi)(1 - 0.33\cos(\psi)), \tag{12}$$

$$E_{\mathrm{Att}}(\phi) = 0.9326(1 - 0.48\cos(\phi) - 0.31\cos(2\phi)), \tag{13}$$

$$F_{\mathrm{Ali}}(d) = \gamma_{\mathrm{Ali}}\left(\frac{d}{d_{\mathrm{Ali}}} + 1\right)\exp\left[-\left(\frac{d}{l_{\mathrm{Ali}}}\right)^{2}\right], \tag{14}$$

$$E_{\mathrm{Ali}}(\psi) = 0.9012[1 + 0.6\cos(\psi) - 0.32\cos(2\psi)], \tag{15}$$

$$O_{\mathrm{Ali}}(\phi) = 1.6385\sin(\phi)(1 + 0.3\cos(2\phi)), \tag{16}$$

where $\gamma_{\mathrm{Att}} = 0.12$ and $\gamma_{\mathrm{Ali}} = 0.09$ are the dimensionless intensity of attraction and alignment respectively, $l_{\mathrm{Att}} = 20$ cm and $l_{\mathrm{Ali}} = 20$ cm are the respective interaction ranges, $d_{\mathrm{Att}} = 3$ cm is the critical distance below which attraction becomes repulsion (preventing collisions between fish whose typical body length is 3 cm), and $d_{\mathrm{Ali}} = 6$ cm.

Fig 2A shows the three most influential neighbors acting on fish *i*. Although fish 2 is close and in front of fish *i*, fish 1, located to the right of fish *i*, has a higher influence on fish *i* ($\mathcal{I}_{i1} > \mathcal{I}_{i2}$) because of the particular shape of the interaction functions: the fish in front is such that $\psi_{i2}$ is small, so $O_{\mathrm{Att}}$ is almost zero (see Fig 2C), while the fish to the right is at $\psi_{i1} \approx 100$ degrees, which is where $O_{\mathrm{Att}}$ reaches its maximum value. As the other state variables are similar, $d_{i1} \approx d_{i2}$ and $\phi_{i1} \approx \phi_{i2}$, the influence is determined by the angle of perception, making fish 1 the most influential fish acting on fish *i*. See S1 Video, showing the identity of the most

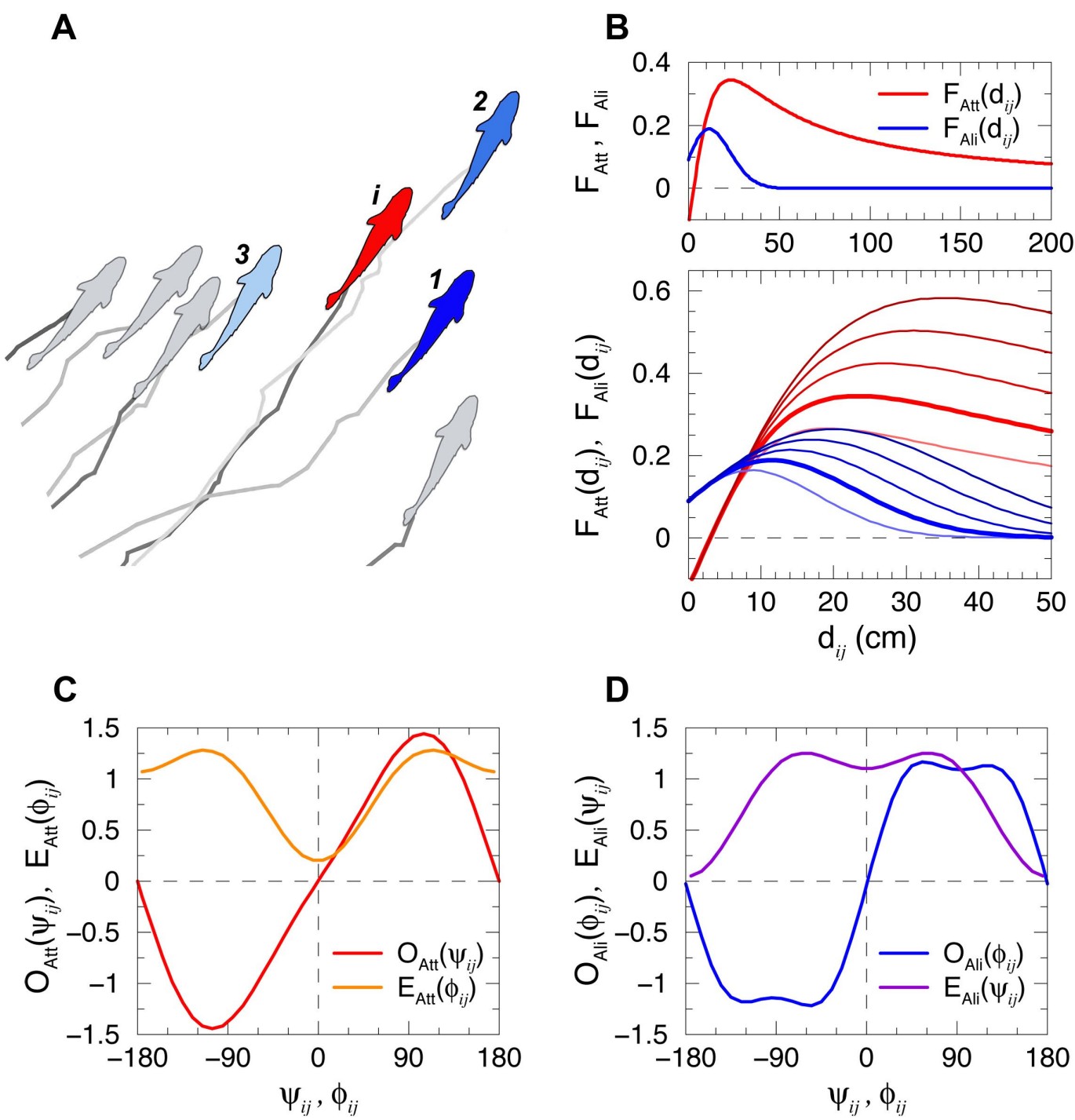

**Fig 2. Social influence and interaction functions between fish.** (A) Example of simulated fish trajectories for a few time steps highlighting the 3 most influential neighbors of fish $i$ (red) according to their respective influence $I_{ij}$ as determined by the model, from the most influential (dark blue) to the least influential one (light blue). (B) Intensity of the attraction $F_{Att}$ (red) and the alignment $F_{Ali}$ (blue) for different interaction ranges $l_{Att}$ = 0.16, 0.2, 0.24, 0.28 and 0.32 m, and $l_{Ali}$ = 0.16, 0.2, 0.24, 0.28 and 0.32 m, from light to dark color intensity. Wider red and blue lines correspond to the values extracted from the experiments with *H. rhodostomus* in [44], $l_{Att}$ = $l_{Ali}$ = 0.2 m. The upper subpanel shows the long range of attraction in $d_{ij} \in [0, 2]$ m. (CD) Normalized angular functions $O_{Att}(\psi_{ij})$ (odd, in red) and $E_{Att}(\phi_{ij})$ (even, in orange) of the attraction interaction, and $O_{Ali}(\phi_{ij})$ (odd, in blue) and $E_{Ali}(\psi_{ij})$ (even, in violet) of the alignment interaction between fish $i$ and $j$, as a function of the angle of perception $\psi_{ij}$ and the relative heading $\phi_{ij}$.

influential fish, identified with respect to a fixed focal fish during swimming dynamics, and S2 Video, with the two most influential neighbors.

## 2.2 Quantification of collective behavior

We characterize the collective patterns by means of three observables quantifying the behavior and the spatial distribution of the school: 1) the group cohesion, measured by the dispersion of individuals with respect to the barycenter of the group, 2) the group polarization, measuring how much fish are aligned, and 3) the milling index, measuring how much the fish turn around their barycenter in a vortex formation [62].

1. Group dispersion $D(t)$ is defined in terms of the distance of each fish to the barycenter of the group, $\|\vec{u}_i - \vec{u}_B\|$:

$$D(t) = \sqrt{\frac{1}{N}\sum_{i=1}^{N} \|\vec{u}_i - \vec{u}_B\|^2}. \tag{17}$$

High values of $D(t)$ correspond to groups of spatially dispersed individuals, while low values of $D(t)$ correspond to highly cohesive groups. Values of $D(t)$ can be arbitrarily high when fish are in a dispersion regime because the distance between fish depends on the duration of the simulation time.

The $x$-coordinates of the position and velocity vectors of the barycenter $B$ are

$$x_B(t) = \frac{1}{N}\sum_{i=1}^{N} x_i(t), \quad v_x^B(t) = \frac{1}{N}\sum_{i=1}^{N} v_x^i(t), \tag{18}$$

with similar expressions for $y_B(t)$ and $v_y^B(t)$. The heading angle of the barycenter is given by its velocity vector, $\phi_B = \text{ATAN2}(v_y^B, v_x^B)$.

2. Group polarization $P(t) \in [0, 1]$ measures the level of alignment of fish independently of the intensity of the speed:

$$P(t) = \frac{1}{N}\left\|\sum_{i=1}^{N} \vec{e}_i(t)\right\|, \tag{19}$$

where $\vec{e}_i = \vec{v}_i/\|\vec{v}_i\| = (\cos\phi_i, \sin\phi_i)$ is the unit vector characterizing fish heading. Values of $P$ close to 1 mean that the $N$ individual headings are aligned and point in the same direction. Values of $P$ close to 0 mean that the $N$ vectors point in different directions, but can also mean that vectors are collinear and with opposite direction so that they cancel each other (*e.g.*, half of the vectors pointing North and the other half pointing South). When headings are uncorrelated, the polarization index is such that $P \sim 1/\sqrt{N}$, which becomes small only for large group size $N$, but which is markedly lower than 1 for any $N \geq 5$. For $N = 100$, a value of $P \approx 0.1$ would mean that the group is not polarized.

3. The milling index $M(t) \in [0, 1]$ measures how much the fish are turning in the same direction around their barycenter $B$,

$$M(t) = \left|\frac{1}{N}\sum_{i=1}^{N} \sin(\bar{\theta}_w^i(t))\right|, \tag{20}$$

where $\bar{\theta}_w^i(t) = \bar{\phi}_i - \bar{\theta}_i$. Variables with a bar are defined in the barycenter system of reference: $\bar{x}_i = x_i - x_B$, $\bar{v}_x^i = v_x^i - v_x^B$ (similar expressions for the $y$-components). Then, the relative

position angle and heading of fish $i$ with respect to $B$ are respectively $\bar{\theta}_i = \texttt{ATAN2}(\bar{y}_i, \bar{x}_i)$ and $\bar{\bar{\phi}}_i = \texttt{ATAN2}(\bar{v}_y^i, \bar{v}_x^i)$. Note that $\bar{\bar{\phi}}_i \neq \phi_i - \phi_B$.

Fish rotating counterclockwise (resp. clockwise) around $B$ contribute to make the sum between bars in Eq 20 to tend to 1 (resp. −1). The milling index $M(t)$ indicates how intense is the group milling, whatever the direction of rotation.

## 2.3 Numerical simulations of the model

For each set of parameter values, we performed 20 simulation runs with different initial conditions over a period of time during which each fish performed 2000 kicks.

In *H. rhodostomus*, the average kick duration being 0.5 s, the total duration of each run corresponds to 16.6 minutes, during which fish are expected to have reached a stable state. Initial transient times (typically 100 s) of each run are not taken into account in the calculation of the average values used in the analysis of the behavioral phases, in order to remove possible effects of the initial condition. These initial transients are however conserved to measure the time it takes to reach a stable state.

The discretization setting to explore the parameter space was chosen to get a detailed visualization of the regions of interest and their evolution from one condition to another (large phase diagrams were obtained with $\Delta\gamma = 2 \times 10^{-3}$, and small phase diagrams with $\Delta\gamma = 10^{-2}$).

At the initial condition (at $t = 0$), all fish are randomly distributed in a circle of radius $R$ chosen so that the mean distance between fish, estimated to be $\approx \sqrt{\pi R^2/N}$, is of the order of half the range of social interactions, ($\approx l_{\text{Att}}/2$, that is, $R = \sqrt{(N/\pi)}l_{\text{Att}}/2$). Initial headings are sampled from the uniform distribution in $(-\pi, \pi)$.

## 3 Results

The data-driven model considered in the present work aims at studying the diversity and plasticity of collective behavior in a specific species of fish (*Hemigrammus rhodostomus*). Yet, the general structure of our burst-and-coast model can be also exploited to describe other species of fish performing intermittent swimming. This is for instance the case of zebra fish (*Danio rerio*) for which the interaction functions have been measured in [45], and where the range of interactions were found to be much shorter, contributing to explain the fact that zebra fish display a weaker collective social behavior than *H. rhodostomus*.

Moreover, within a model for a specific species, it is certainly worthwhile to study the behavior of the model when changing the value of its parameters (intensity and range of the interactions, intensity of the cognitive noise), and in particular, to address the possible occurrence of non-trivial collective states. Not only this is relevant on a theoretical point of view, but it is also expected that actual fish can experience different effective interaction parameters depending on their age, activity, or their environment. For instance, it is shown in [31] that the intensity of the alignment interaction for *Kuhlia mugil* is roughly proportional to the mean velocity of the school. Moreover, water turbidity [46–48] and/or light intensity [49] have been shown to impact the interactions between fish. For instance, the authors of [49] have shown that in darkness and under low illuminance, *H. rhodostomus* move with lower polarization and a large mean nearest-neighbor distance that can span many body lengths. In the same work, the authors showed that it is possible to change the intensity of the attraction between fish by compromising the lateral-line system with a specific antibiotic treatment.

Hence, in the following, we explore the parameter space of the *H. rhodostomus* model by varying its main parameters, which are the intensity of the attraction and alignment

interactions $\gamma_{Att}$ and $\gamma_{Ali}$, their respective range $l_{Att}$ and $l_{Ali}$, the intensity of the random heading fluctuation $\gamma_R$ (cognitive noise), and the group size $N$. Results are presented in the form of two-dimensional phase diagrams obtained by varying the intensity of the attraction and alignment interactions, $\gamma_{Att}$ and $\gamma_{Ali}$, while keeping other parameters fixed (interaction ranges, cognitive noise, group size).

We first present the most illustrative cases where the regions corresponding to the different phases of collective behaviors can be easily identified. We then describe the impact of the other parameters on these characteristic collective phases.

In each case, two social interaction strategies are considered, each one determined by the $k$ selected neighbors (with $k = 1$ and $k = 2$) that have the largest influence on the focal fish when it performs a new kick.

### 3.1 Characterization and analysis of the collective states

Fig 3 shows the color maps of dispersion, polarization, and milling as a function of the parameters $\gamma_{Att}$ and $\gamma_{Ali}$ where significative variations have been found, for the two social strategies $k = 1$ and $k = 2$, in the particular case where the social interaction ranges are $l_{Att} = l_{Ali} = 0.28$ m, the intensity of random fluctuations (noise) is $\gamma_R = 0.2$, and the group size is $N = 100$. For $k = 1$, variations take place for $(\gamma_{Att}, \gamma_{Ali})$ in $[0, 0.1] \times [0, 0.6]$, and for $(\gamma_{Att}, \gamma_{Ali})$ in $[0, 0.1] \times [0, 0.4]$ for $k = 2$.

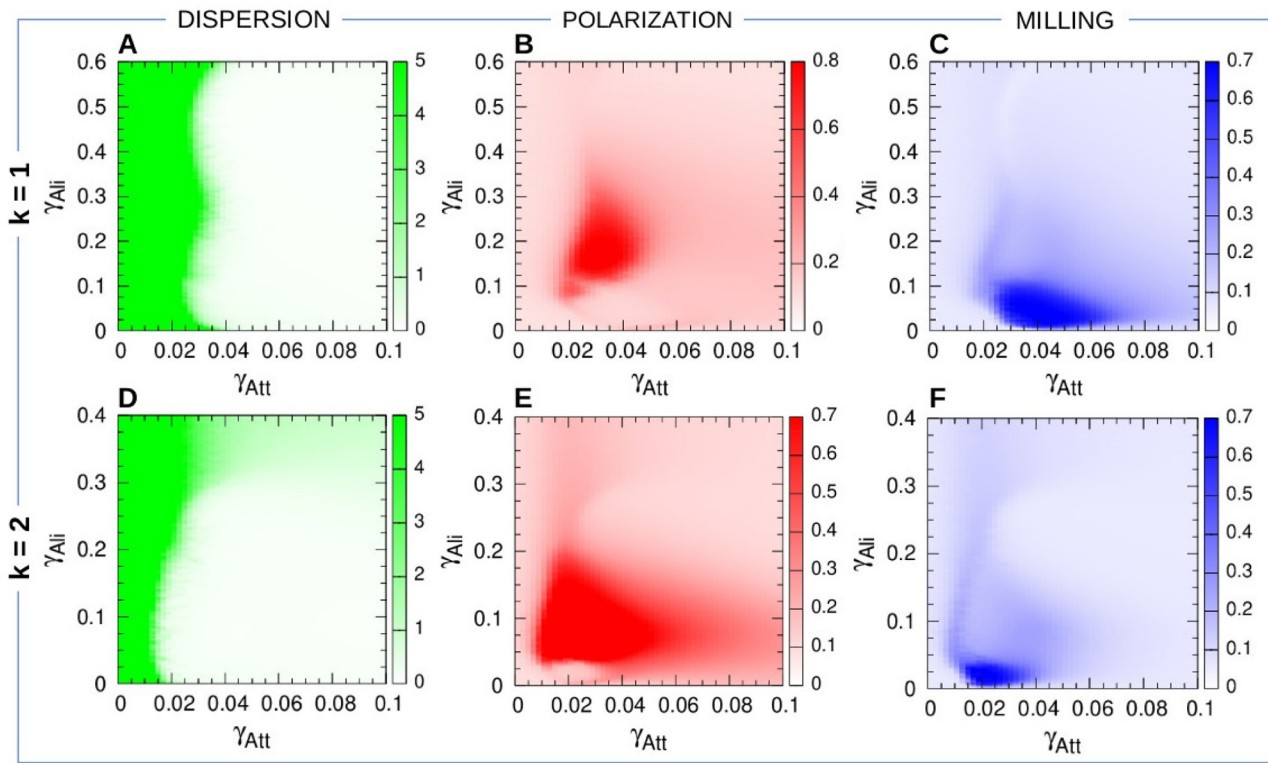

**Fig 3. Dispersion, polarization, and milling as functions of the intensity of attraction $\gamma_{Att}$ and alignment $\gamma_{Ali}$ when fish interact with their most influential neighbor ($k = 1$) and their two most influential neighbors $k = 2$.** (AD) Dispersion: color intensity is proportional to dispersion. White region means that the school of fish is highly cohesive. In the green region, attraction is too weak and fish quickly disperse. (BE) Polarization. (CF) Milling. Regions of high color intensity mean that fish frequently display either the characteristic behavioral patterns of schooling or milling. Social interaction ranges are $l_{Att} = l_{Ali} = 0.28$ m, intensity of random fluctuation (noise) is $\gamma_R = 0.2$. Each pixel is the average of 20 runs of 2000 kicks per fish (in which the first 100 kicks have been discarded).

The first remarkable result is that, despite the discrete character of the burst-and-coast swimming mode, the group of fish remains cohesive in a large region of the parameter space, even when fish interact only with their most influential neighbor, as testified by the wide regions in Fig 3A and 3D. For both interaction strategies $k = 1$ and $k = 2$, the dispersion map consists of two regions of high and low values of $D$ separated by a thin transition layer. In the green region, $\gamma_{Att}$ is too small and the attraction is not strong enough to preserve the cohesion of the group and fish disperse. The high value shown in the green regions of Fig 3A and 3D (higher than 5) is determined by the duration of the numerical simulations; longer runs give rise to higher values of $D(t)$. This is not the case in the white region, where the group remains cohesive whatever the length of the simulation time.

The transition layer is almost vertical and very thin (width $\approx 0.005$). Its location is mostly determined by the value of $\gamma_{Att}$, which is of about 0.03 when $k = 1$ and of about 0.015 when $k = 2$, showing that cohesion is mostly mediated by attraction, alignment having only a marginal contribution. When $k = 2$, an horizontal light green region where $D \approx 1$ is visible in Panel D when $\gamma_{Ali} > 0.3$. In this region, the value of $\gamma_{Ali}$ is so large that the contribution of the alignment to the heading angle change $\delta\phi$ in Eq 4 makes the fish to rotate an angle often larger than $\pi$, as if the new heading angle was chosen randomly. For these values, the fish behave more or less like slow random walkers. This region of slow dispersion is not visible when $k = 1$ because in this case Eq 4 has only one contribution to the increase of $\delta\phi$. The same kind of dispersion is observed when $\gamma_{Att}$ is very large, for both $k = 1$ and $k = 2$. We have tried to determine the onset of this region in the interval of values of $\gamma_{Att}$ for $k = 1$. Increasing the value of $\gamma_{Att}$ produces a stronger attraction and should in principle give rise to a more cohesive group. However, we found that an optimal value of the attraction strength exists, $\gamma_{Att}^{*} \approx 0.5$, for which the distance to the nearest neighbor is minimal. This means that, above $\gamma_{Att}^{*}$, increasing $\gamma_{Att}$ leads to an increase in the distance to the nearest neighbor, which is a clear indicator that cohesion begins to weaken. We have calculated this optimal value for $\gamma_{Ali} \in [0, 0.3]$, finding that it is more or less the same.

The second remarkable result is that significantly high values of polarization ($P > 0.5$ in Fig 3B and 3E and milling ($M > 0.4$ in Fig 3C and 3F) are observed in wide regions of the parameter space. This result shows that groups of fish are able to display the characteristic collective behaviors of schooling and milling, even if each fish only interacts with their most influential neighbor. Both the schooling and milling regions are mostly contained in the region of high cohesion, although the schooling region seems to slightly overlap with the dispersion region when $\gamma_{Att}$ is small. This corresponds to situations where the group is initially aligned (attraction is weak, so interaction is mostly alignment) and then splits in several subgroups. Although fish move away from each other, they remain more or less aligned inside each subgroup and also with respect to other subgroups, due to the persistence of the initial alignment, thus contributing to produce high values of $P(t)$. As mentioned above for the region of dispersion, longer simulations would give rise to a lost of this persistence and thus to the shrinking of the region of high polarization, reducing it to the points contained in the white region of dispersion in Fig 3A.

We performed a series of very long simulations of our model for values of $\gamma_{Att}$ and $\gamma_{Ali}$ across the region of high polarization in Fig 3B, that is, for $\gamma_{Att}$ going from 0.026 to 0.06 and for a fixed value of $\gamma_{Ali} = 0.2$ (20 times longer simulations, until $t = 20.000$ s, each point is the average of 100 runs). S4 Fig shows that the polarization is maximal and remains at its highest value $P \approx 0.85$ for a small interval of the attraction strength $\gamma_{Att} \in [0.034, 0.04]$. Outside this interval, high polarization is lost, and this occurs in two different ways, depending on if $\gamma_{Att}$ is smaller or higher than the values contained in this interval. For values of $\gamma_{Att}$ below 0.034, the

polarization decreases because the attraction strength is so small that the group splits in subgroups (S5 Fig). Even if fish may be still aligned inside each subgroup, they are not necessarily aligned with fish from other subgroups, and consequently the polarization of the total group decreases. For values of $\gamma_{Att}$ higher than 0.04, the attraction strength is sufficiently high to maintain the group cohesive (there is only one group, as shown in S5 Fig). In turn, this strong attraction makes the fish to rotate excessively towards their neighbor, thus unbalancing the alignment between fish and consequently the polarization of the group. This effect is stronger the higher $\gamma_{Att}$ is, and is a direct consequence of the discrete nature of the model. For larger values of $\gamma_{Att}$, above 0.06, the polarization has disappeared and the group is in a swarming state.

The transition from high to low polarization as $\gamma_{Att}$ decreases depends on the duration of the simulations. Each line in S4 Fig represents more than 5 hours of real swimming of the whole fish group, a time during which environmental conditions are expected to vary considerably, at least in light and temperature. In Fig 3 (and the rest of simulations in the paper), each point is the result of simulations of $\sim 16.6$ minutes of swimming time, a reasonable time duration during which changes in environmental conditions are considered as not affecting fish behavior.

The schooling region also overlaps with the milling region, meaning that both $P$ and $M$ have high average values for the same combination of parameters. This corresponds to situations where the group alternates between schooling and milling, and where wide turns (of radius equal to several times the radius of the group) can also be observed.

In order to analyze the relative size and location of the collective phases and to describe the impact of the interaction strategies and the other parameters on these behavioral phases, we synthesize the information in a phase diagram that puts together the distinct behavioral phases (see Fig 3).

We define the following four phases of collective behavior:

I). $P \geq 0.4$ and $M \leq 0.4$ correspond to the SCHOOLING phase, in red.

II). $P \leq 0.4$ and $M \geq 0.4$ correspond to the MILLING phase, in blue.

I-II). $P > 0.4$ and $M > 0.4$ correspond to the INTERMITTENT region, in cyan.

III). $P < 0.4$ and $M < 0.4$ correspond to the SWARMING phase, in green.

The typical behavioral patterns displayed by the group in each of the four phases are illustrated in Fig 4A and in S3–S7 Videos.

Fig 4 shows the resulting phase diagram corresponding to each strategy in Panels B and C, with the borders of the regions overlaying the dispersion map in Panels D and E. Phase diagrams are centered on the region of interest, $(\gamma_{Att}, \gamma_{Ali}) \in [0, 0.08] \times [0, 0.4]$, using the same scale for both strategies $k = 1$ and $k = 2$.

Fig 4B shows that the SCHOOLING and MILLING phases are precisely located in the zone between the region of high dispersion (gray region) and the region of high cohesion (green). When attraction is too small, fish are dispersed. When the attraction is larger than $\gamma_{Att} \approx 0.03$ for $k = 1$ (Fig 4D) and $\gamma_{Att} \approx 0.015$ for $k = 2$ (Fig 4E), fish reaction to its neighbor becomes sufficiently strong to coordinate collective movements, whose particular form depends on the intensity of the alignment.

Let us describe first the case $k = 1$ for growing values of $\gamma_{Att}$. For $\gamma_{Att} \approx 0.03$ and $0.01 < \gamma_{Ali} < 0.1$, the fish turn around their barycenter in a typical milling formation shown in Fig 4A; for $0.12 < \gamma_{Ali} < 0.3$, the strength of the alignment induces the fish to swim almost all in the same direction, producing a fish schooling formation (Fig 4A(I)). For higher values of the alignment

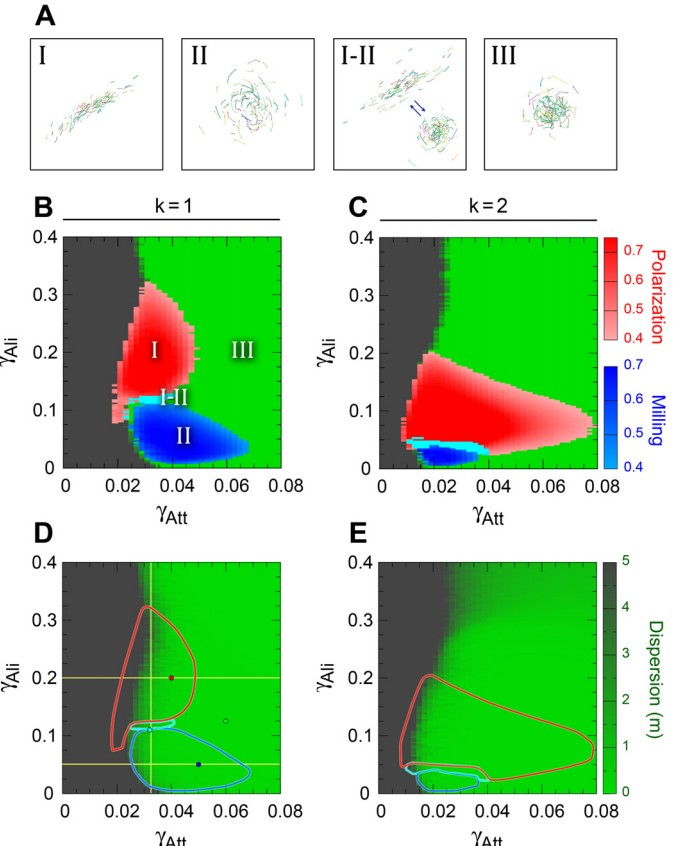

**Fig 4. Impact of interaction strategies on collective states.** (A) Spatial configuration of the characteristic collective states displayed by a group of $N = 100$ fish when each fish interacts with its most influential neighbor ($k = 1$). (BC) Phase diagrams showing four distinct behavioral regions: SCHOOLING (I) in red, with highly polarized group ($P > 0.4$, $M < 0.4$), MILLING (II) in blue, with low polarized group frequently displaying milling behavior ($P < 0.4$, $M > 0.4$), SWARMING (III) in green, with small polarization and milling ($P < 0.4$, $M < 0.4$), intermittence between schooling and milling (I-II) in cyan ($P > 0.4$, $M > 0.4$) when fish interact with their most influential neighbor (B) and their two most influential neighbors (C). Gray regions correspond to excessively weak or excessively strong interactions, giving rise to fish dispersion ($D > 5$ m). Boundary between high dispersion region and swarming region (both in uniform color) is determined by the transition layer from high to low dispersion shown in Panels D and E. (DE): Contour lines of the collective behavior phases shown in Panels (BC), overlaying the dispersion map, from green (low dispersion) to gray (high dispersion). In panel (D), small circles denote the combination of parameters ($\gamma_{Att}$, $\gamma_{Ali}$) corresponding to the time series shown in Fig 5, and yellow lines denote the vertical and horizontal cuts described in Fig 6. Social interaction ranges are $l_{Att} = l_{Ali} = 0.28$ m, intensity of random fluctuation (noise) is $\gamma_R = 0.2$.

strength (*i.e.*, when $\gamma_{Ali} > 0.3$), the school is disorganized and forms a swarm (Fig 4A(III)). Indeed, because of the discrete nature of the model, when the intensity of the alignment becomes too strong, the heading change of the focal fish becomes too large and effectively random, which leads to a decrease in the group polarization and even to a dispersion of the group for $\gamma_{Ali}$ of order 1.

Larger values of $\gamma_{Ali}$, not appearing in the phase diagram, correspond to the transition to the slow dispersion region described above. For given values of $\gamma_{Ali}$ in the narrow interval [0.12, 0.13], the group alternates milling and schooling along time. Intermediate states appear, in which fish are more or less aligned, forming a curved group that describes wide turns of a radius larger than several times the width of the group when it is in a milling formation. In both phases, these intervals of $\gamma_{Ali}$ become smaller as larger values of $\gamma_{Att}$ are considered. Thus,

the SCHOOLING phase lasts until $\gamma_{Att} \approx 0.05$ for $\gamma_{Ali} \approx 0.2$, and the MILLING phase goes up to $\gamma_{Att} \approx 0.065$ for $\gamma_{Ali} \approx 0.05$. The transition from schooling and milling phases to the swarming phase results from the unbalance of attraction and alignment interactions. For instance, when attraction is too strong, or, equivalently, alignment is too weak, fish turn excessively towards their neighbors, in detriment of the alignment required to preserve group configuration. For larger values of $\gamma_{Att}$, not appearing in the phase diagram, we find a region of slow dispersion where fish move as if their heading angle varies in a random way.

Extending the number of interacting neighbors from $k = 1$ to 2 strongly reduces the size of the MILLING phase, which shrinks in width from about 0.045 to about 0.022 and in height from 0.1 to 0.026 (Fig 4C). This is due to the fact that fish now pay attention to two neighbors, thus destabilizing the milling formation, in benefit of the schooling formation, which absorbs most of the milling region. The SCHOOLING phase is indeed much wider than when $k = 1$, growing in width from 0.02 to 0.065, for almost the same height of about 0.15, but for a smaller range of the alignment: when $k = 1$, schooling takes place for $\gamma_{Ali} \in [0.125, 0.3]$, while when $k = 2$, schooling appears when $\gamma_{Ali} \in [0.025, 0.2]$.

In order to understand how the transition between collective states take place, we explored in detail the time series of the three observables $D(t)$, $P(t)$, and $M(t)$, in a number of representative cases. Fig 5 shows the time series of polarization (red) and milling (blue) for the four sets of parameters $(\gamma_{Att}, \gamma_{Ali})$ represented in Fig 4D, one for each distinct behavioral phase. Panels A, B, and D of Fig 5 clearly correspond to what is expected from the phase diagram. Note that these time series are 5 times longer than those used to calculate the average values plotted in

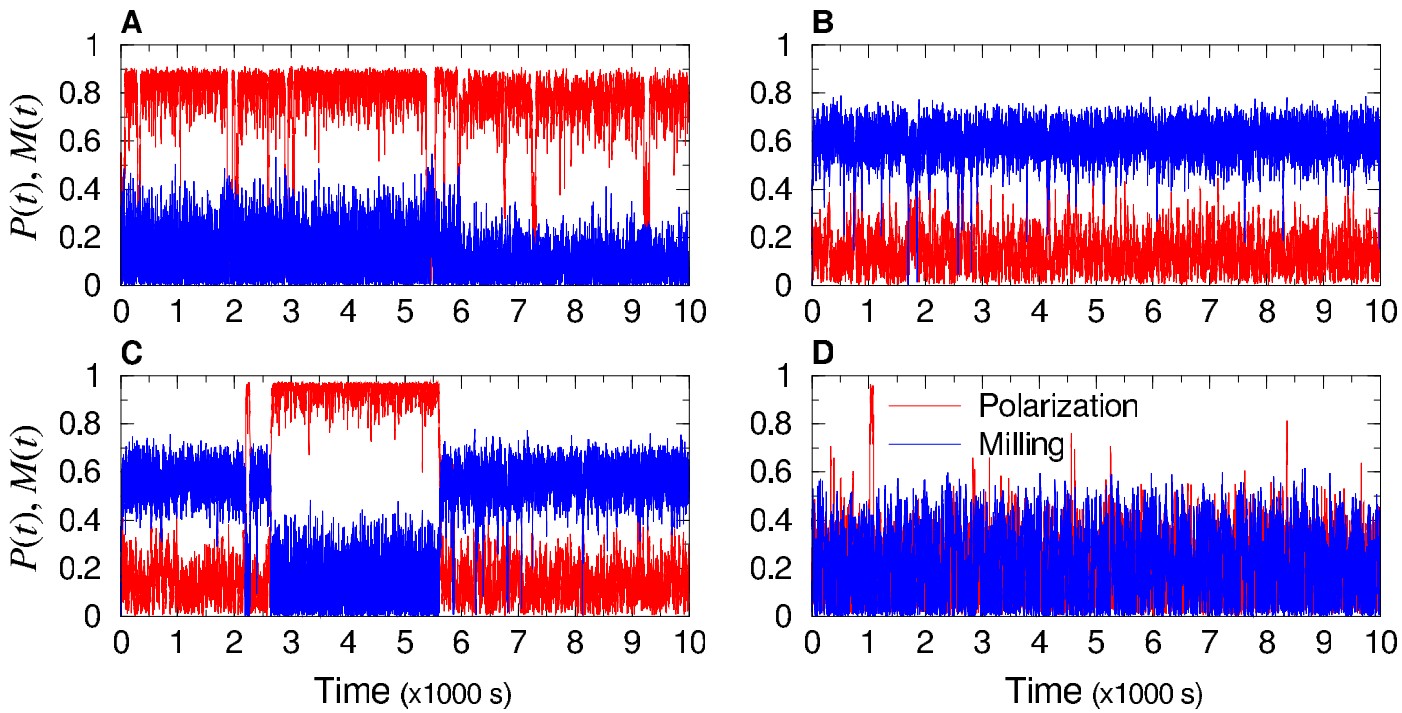

**Fig 5. Time series of polarization (red line) and milling (blue line), representative of each characteristic collective state by a group of $N = 100$ fish when each fish only interacts with its most influential neighbor ($k = 1$).** (A) SCHOOLING state: $(\gamma_{Att}, \gamma_{Ali}) = (0.04, 0.2)$, high polarization $P(t) \approx 0.9$, low milling $M(t) \approx 0.22$. (B) MILLING state: $(\gamma_{Att}, \gamma_{Ali}) = (0.05, 0.05)$, low polarization $P(t) \approx 0.16$, high milling $M(t) \approx 0.6$. (C) Intermittent state: $(\gamma_{Att}, \gamma_{Ali}) = (0.033, 0.11)$, region where high polarization and high milling alternate: $P(t) \approx 0.98$ with $M(t) \approx 0.22$, and $P(t) \approx 0.18$ with $M(t) \approx 0.6$. (D) SWARMING state: $(\gamma_{Att}, \gamma_{Ali}) = (0.06, 0.125)$, low values of both polarization and milling: $P(t) \approx 0.2$, $M(t) \approx 0.2$. Social interaction ranges are $l_{Att} = l_{Ali} = 0.28$ m, intensity of random fluctuation (noise) is $\gamma_R = 0.2$. Time interval is $[0, 10000]$ s, i.e., more than 2.5 hours.

the phase diagram. Less obvious is what happens in the transition region between schooling and milling. Panel C indicates that different collective states can emerge with the same set of parameters, and that the averaging process used to draw the phase diagram can hide complex combinations of patterns.

In order to explore how frequent these patterns of bistability are, we computed the probability density functions (PDF) of polarization and milling along three cuts crossing the identified phases regions in Fig 4D (yellow lines): one vertical cut at $\gamma_{\text{Att}} = 0.0325$, and two horizontal cuts at $\gamma_{\text{Ali}} = 0.2$ and $\gamma_{\text{Ali}} = 0.05$. The PDFs are calculated over the 20 runs of 2000 kicks per fish, initial transient excluded.

Fig 6 shows the resulting PDFs, with the mean value depicted in the phase diagram superimposed to each PDF. Panels A and B show respectively the PDF of polarization and milling along the vertical cut for $\gamma_{\text{Ali}} \in [0, 0.4]$. Each phase is clearly recognizable: MILLING phase for $0.02 < \gamma_{\text{Ali}} < 0.08$ (low values of $P$ below 0.15, high values of $M$ above 0.5 and up to 0.8), SCHOOLING phase for $0.14 < \gamma_{\text{Ali}} < 0.28$ (high values of $P$ above 0.5 and up to 0.9, low values of $M$ below 0.18), and SWARMING phase for $\gamma_{\text{Ali}} < 0.02$ and $\gamma_{\text{Ali}} > 0.3$ (values of $P$ below 0.5 and values of $M$ below 0.2). One can notice a first sharp transition to milling state at $\gamma_{\text{Ali}} \approx 0.02$, then milling decreases until $\gamma_{\text{Ali}} \approx 0.1$, and then there is another sharp transition to schooling state at $\gamma_{\text{Ali}} \approx 0.12$.

In the SCHOOLING phase, the level of polarization is smaller for higher values of the alignment strength. This is a consequence of the higher amplitude that angular changes have in the swimming direction of fish when $\gamma_{\text{Ali}}$ is high, thus decreasing the group polarization. This effect, which at first glance may appear paradoxical, reveals the impact on collective behaviors of a burst-and-coast swimming mode in which there is a discontinuous influence of social interactions on the direction of swimming of fish.

For larger values of the alignment strength (*i.e.*, when $\gamma_{\text{Ali}} > 0.3$), the heading angle change $\delta\phi$ is so large that fish turn excessively and can hardly synchronize their movements with their neighbors, making the group to adopt a swarming phase.

Of particular interest is the intermediate region where $0.08 < \gamma_{\text{Ali}} < 0.14$ (see Fig 6A and 6B). There, the PDFs of both $P$ and $M$ exhibit two peaks, thus revealing the alternation between the high polarization and high milling intervals, as shown in the time series in Fig 5C. Moreover, the polarization is higher in this intermittent region ($P \approx 0.95$) than in the pure schooling phase ($P \approx 0.85$).

Phases are equally easily identifiable in the horizontal cuts (Fig 6C–6F). No ordered phases appear when the strength of attraction is too small (*i.e.*, when $\gamma_{\text{Att}} < 0.02$), for both values of $\gamma_{\text{Ali}}$. The SCHOOLING phase shows a high value of $P \approx 0.85$ from $\gamma_{\text{Att}} \approx 0.028$ to 0.048, where the PDFs start to exhibit a second wide peak at low values of $P \approx 0.2$, corresponding to the transition from the schooling phase to the SWARMING phase. The MILLING phase, crossed by the lower horizontal cut, starts at a higher value of $\gamma_{\text{Att}} \approx 0.032$, and spans a wider interval up to $\gamma_{\text{Att}} \approx 0.064$, where, as for the schooling phase, the transition to the SWARMING phase starts.

It is remarkable that cohesive and even ordered (schooling; milling) collective phases are obtained when fish only interact with their most influential neighbor ($k = 1$), whereas only interacting with the nearest neighbor would not even permit the cohesiveness of the school, as shown in [60]. In S6 and S7 Figs, we respectively show the PDF of the number of different most influential neighbors (DMIN) and of the number of different first nearest neighbors (DFNN), at any given time, for a school of $N = 100$ fish, and for interaction parameters chosen deep in the 3 different collective phases (swarming; schooling; milling). Note that for a physical attractive interaction only depending on the distance between interacting particles (like gravity), DFNN would exactly coincide with the DMIN. We find that the mean number of DMIN (39 for schooling and swarming states; 46 for the milling state) is very significantly smaller

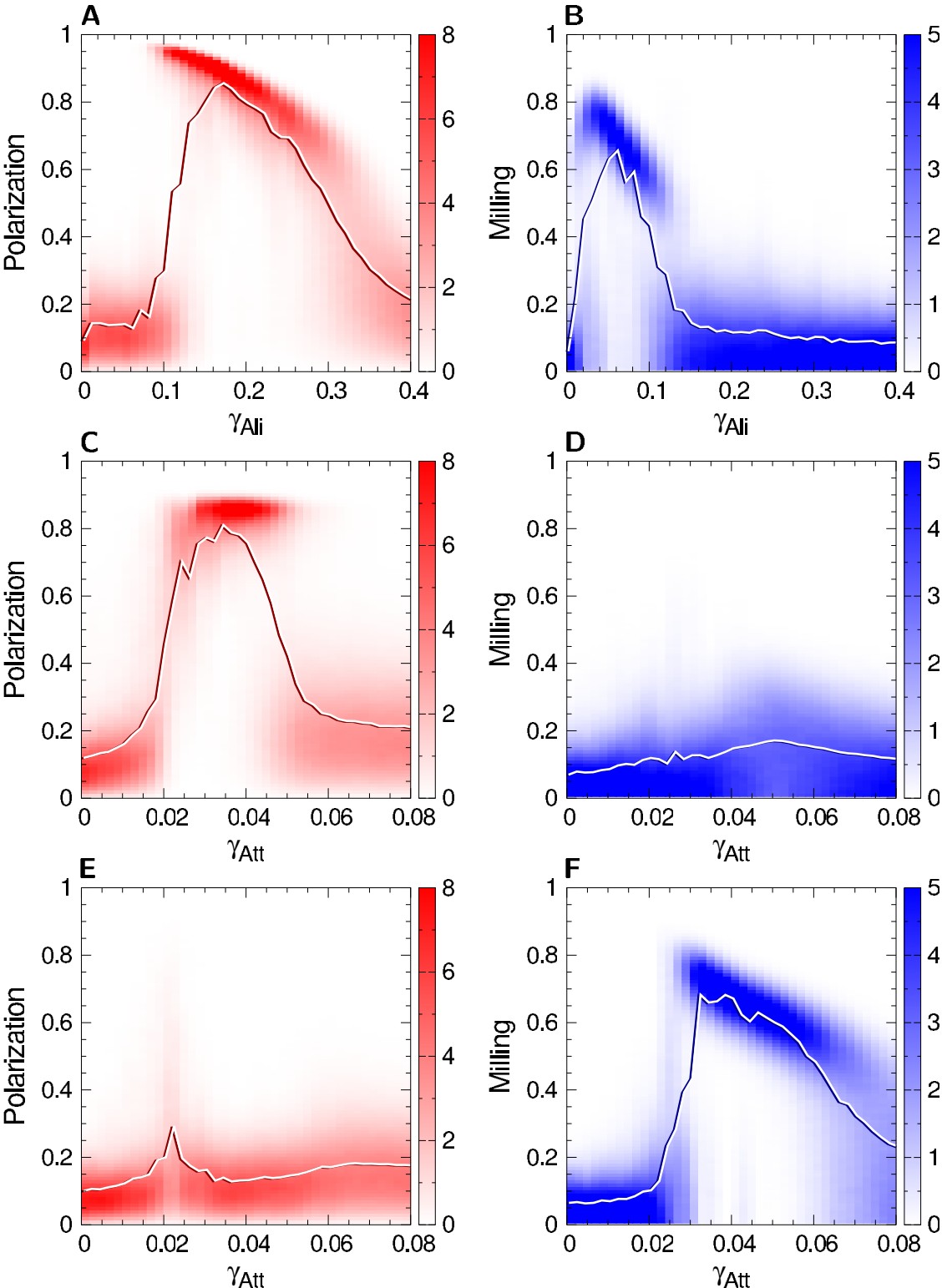

**Fig 6. Polarization (red) and milling (blue) probability distribution functions and mean values (solid lines) along one vertical and two horizontal cuts of the phase diagram for the case *k* = 1.** (AB) Vertical cut at $\gamma_{Att}$ = 0.0325, (CD) Horizontal cut at $\gamma_{Ali}$ = 0.2, and (EF) Horizontal cut at $\gamma_{Ali}$ = 0.05. Each vertical line is the probability distribution function (PDF) of the corresponding value of the parameter shown in the *x*-axis, for the parameter of each figure. Color intensity is proportional to the frequence of a value in the vertical axis. Solid lines are the corresponding mean value of each PDF. Interaction ranges are $l_{Att} = l_{Ali}$ = 0.28 m and noise intensity is $\gamma_R$ = 0.2.

than the number of DFNN (mean 71 in the 3 collective phases), showing that a typical DMIN is linked (and interacts) with more fish than a DFNN (2.2–2.6 fish linked to each DMIN; 1.4 fish linked to each DFNN). Therefore, the dynamical notion of most influential neighbors is more effective at propagating information within the school, and hence ensuring cohesiveness and a possible long-range order (schooling and milling). Interestingly, the milling phase is characterized by more DMIN than the 2 other phases (which are indiscernible under this criterion), possibly because more information exchange within the school is needed to ensure the cohesiveness, alignment, *and* collective rotation of the school.

In summary, small variations of the parameter values that control the intensity of attraction and alignment lead to sharp changes in collective states that emerge at the group level. Furthermore, each of the two parameters $\gamma_{Att}$ and $\gamma_{Ali}$ can independently control the emergence of the schooling and milling phases in certain ranges of their value. In other words, both attraction and alignment interactions can have a synergistic effect on the stability of the different collective states. In addition, and confirming some results obtained in [60], the notion of most influential neighbors (compared to the more naive notion of nearest neighbors) is crucial to ensure the existence of cohesive and even ordered states within interacting rules only involving a very few fish (1 or 2). As already discussed in [60], the biological relevance of this notion of most influential neighbors could emerge from cognitive processes filtering the environment of real fish.

### 3.2 Impact of interaction ranges and individual behavioral fluctuations on the collective states

The typical range of action of social interactions and the amplitude of fluctuations in the spontaneous behavior of fish are key parameters that affect the collective state of the group. For given values of $\gamma_{Att}$ and $\gamma_{Ali}$, both the module and the range of the social interaction functions $F_{Att}$ and $F_{Ali}$ depend on the respective parameters $l_{Att}$ and $l_{Ali}$ (see Fig 2B). We consider the case where $l_{Att} = l_{Ali}$, and refer to the interaction ranges by means of the single variable $l_{Att}$. The intensity of $\gamma_R$ reflects the amplitude of spontaneous decisions of the fish to change its heading; intuitively, the group will tend to be more coordinated with smaller values of $\gamma_R$, *i.e.*, high values of spontaneous fluctuations in the individual behavior will counterbalance the effect of social interaction and the ability of the group to self-organize into ordered phases.

We thus analyze the impact that the interaction range and the fluctuations in the spontaneous behavior on the relative size and location of the collective states for both social interaction strategies. In the previous section, we focused on the particular case where $l_{Att} = l_{Ali} = 0.28$ m and $\gamma_R = 0.2$, where collective phases are clearly recognizable and have similar size. Experimental data corresponding to pairs of *H. rhodostomus* swimming in a circular tank have close but slightly different values, $l_{Att} = l_{Ali} = 0.2$ m and $\gamma_R = 0.45$ [44]. In larger groups, fish have a strong tendency to conform to the speed and direction of motion of the group, so that the individual spontaneous fluctuations of heading direction have a smaller amplitude [40, 51]. We thus present the results for five values of $l_{Att}$ and $l_{Ali}$ around the value found for pairs of *H. rhodostomus*, 0.16, 0.2, 0.24, 0.28 and 0.32, for the value of $\gamma_R = 0.45$ found for *H. rhodostomus* and an additional intermediate value $\gamma_R = 0.3$.

Fig 7 shows the combined effects of $l_{Att}$ and $\gamma_R$ on collective states when fish interact with their $k = 1$ or $k = 2$ most influential neighbors.

As expected, increasing the amplitude of spontaneous fluctuation in the individual behavior (cognitive noise) reduces the extent of the ordered phases (schooling and milling) and even leads to their complete disappearance, for all values of $l_{Att}$ and both $k = 1$ and $k = 2$. The MILL-ING phase shrinks and disappears whatever the value of $l_{Att}$ in Fig 7 as $\gamma_R$ grows for both $k = 1$

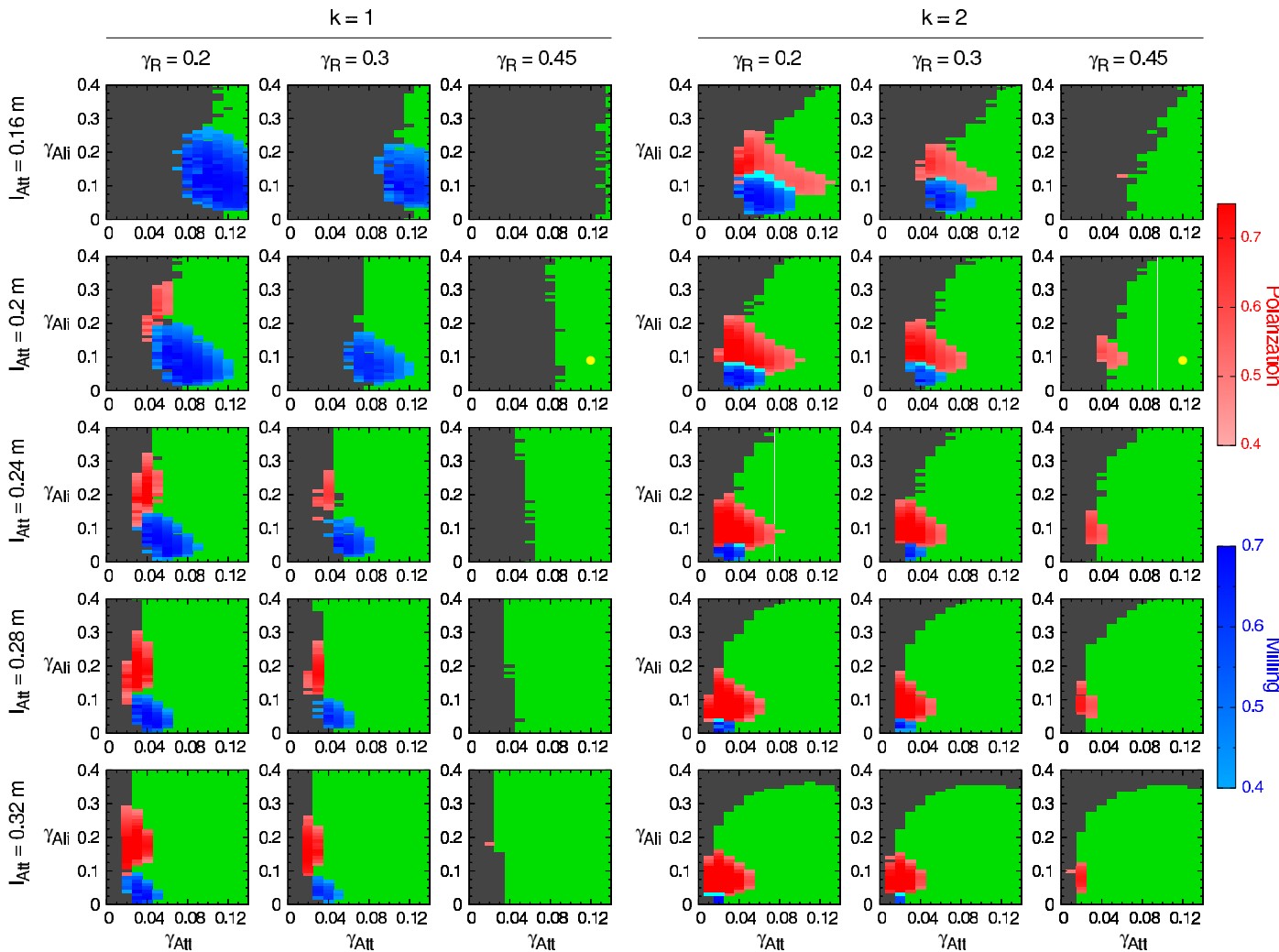

**Fig 7. Phase diagrams for $k = 1$ and 2 for different combinations of the interaction ranges $l_{Att} = l_{Ali}$ and the intensity of random fluctuations $\gamma_R$.** Each row corresponds to a given value of the interaction range; from top to bottom: $l_{Att} = l_{Ali} = 0.16$, 0.2, 0.24, 0.28, and 0.32 m. Each column corresponds to a given value of random fluctuations; from left to right: $\gamma_R = 0.2$, 0.3, and 0.45. Color phases are SCHOOLING in red ($P > 0.4$), MILLING in blue ($M > 0.4$), and SWARMING in green. Cyan color corresponds to $P > 0.4$ and $M > 0.4$. Gray color corresponds to region of high dispersion ($D > 5$ m). Values estimated from experiments with pairs of *H. rhodostomus* swimming in a small circular tank are $l_{Att} = l_{Ali} = 0.2$ m and $\gamma_R = 0.45$, marked with a yellow dot in the second row and columns 1 ($k = 1$) and 4 ($k = 2$). We used a fine discretization with $\Delta\gamma = 10^{-2}$ on both axes.

and $k = 2$, and only a small SCHOOLING phase remains for $k = 2$ when $\gamma_R = 0.45$. High noise values at the individual scale also lead to the shrinking of the SWARMING phase when $k = 1$, although this effect weakens for larger values of $l_{Att}$ and is only slightly perceptible when $k = 2$.

As observed in the previous section, paying attention to the second most influential neighbor contributes to the shrinking of the MILLING phase, in benefit of a wider SCHOOLING phase. For each combination of $l_{Att}$ and $\gamma_R$, the blue region gets smaller and the red region larger when $k = 2$.

Increasing the ranges of social interaction ($l_{Att}$ and $l_{Ali}$) amounts first to increase the intensity of the attraction. This effect is clearly visible in the reduction of the dispersion region (figured in gray): fish remain cohesive for a wider range of values of $\gamma_{Att}$; see, *e.g.*, column 1 for $k = 1$ and $\gamma_R = 0.2$, where the transition layer between the dispersive and swarming regions

moves to the left from $\gamma_{\text{Att}} \approx 0.13$ for $l_{\text{Att}} = 0.16$ m to $\gamma_{\text{Att}} \approx 0.03$ for $l_{\text{Att}} = 0.32$ m. See S1 Fig for the continuous dispersion map detailing this transition layer for the same conditions. When $k = 1$, only the transition towards small values of $\gamma_{\text{Att}}$ appears in the depicted portion of the phase planes.

For $k = 2$, the same effect is visible in the transition layer when $\gamma_{\text{Ali}}$ is smaller than 0.3, which is the region where the group is still cohesive; see Fig 4E, where higher values of the dispersion (represented by a darker shadow of green) appears precisely above $\gamma_{\text{Ali}} = 0.3$, and columns 4, 5, and 6 in S1 Fig. For the higher values of $l_{\text{Att}}$, in the upper region of the phase diagram, the dispersion dominates because the combined effect of attraction and alignment to the heading angle change $\delta\phi$ in Eq 4 makes the fish to rotate randomly so that they move as random walkers, thus breaking any residual coordination in the fish movements. S1 Fig shows clearly the wide transition from swarming to dispersion in the upper-right part of the phase planes, especially when $l_{\text{Att}}$, $l_{\text{Ali}}$, and $\gamma_{\text{R}}$ are large.

The shift to the left of the transition layer as $l_{\text{Att}}$ grows is accompanied by the same shift to the left of the SCHOOLING and MILLING phases. As explained in the previous section, increasing the attraction strength reduces the range of milling, so increasing $l_{\text{Att}}$ has the same effect, clearly visible in, *e.g.*, column 1 for $k = 1$, where the height of the MILLING phase decreases from 0.25 (at $\gamma_{\text{Att}} \approx 0.11$) to less than 0.1 (at $\gamma_{\text{Att}} \approx 0.03$). This effect is visible for all values of $\gamma_{\text{R}}$ and both $k = 1$ and 2.

The same effect is also found in the transition from the SCHOOLING phase to the SWARMING phase as $\gamma_{\text{Att}}$ grows, that is, to the right of the red region. Note that, for these phase diagrams, increasing $l_{\text{Att}}$ also means that $l_{\text{Ali}}$ increases the same way, so the alignment force becomes stronger; see column 4, for $k = 2$, where the width of the red phase decreases from 0.1 (at $\gamma_{\text{Ali}} \approx 0.1$) to 0.04 (at $\gamma_{\text{Ali}} \approx 0.07$).

Larger values of $l_{\text{Att}}$ also amounts to larger interaction ranges (both in attraction and alignment). The impact of this effect is however more difficult to perceive on the phase diagrams.

One could think that increasing the interaction range in a milling state would imply that fish can pay attention to neighbors that are further away but have a relative orientation that induces a stronger influence, so that the milling state turns into the schooling state. Finally, when $k = 2$, for the same value of the amplitude of the individual fluctuations, the increase in the interaction range reduces the areas of the parameter space that lead to ordered phases. The extent of the schooling and milling phases decreases as $l_{\text{Att}}$ and $l_{\text{Ali}}$ increase, and these collective states only exist for low values of attraction and alignment.

## 3.3 Impact of group size on the collective states

In this section, we investigate the impact of the group size $N$ on the collective states. We perform a series of simulations for $N = 25, 50, 100,$ and 200 fish at fixed $l_{\text{Att}} = l_{\text{Ali}} = 0.28$ m and $\gamma_{\text{R}} = 0.2$ for both $k = 1$ and $k = 2$, the same values used for the illustrative case shown in Figs 3 to 6.

Fig 8 shows the phase diagrams corresponding to these conditions. The first observation is that both the SCHOOLING and MILLING phases appear for all group sizes and social interaction strategies. In all cases, the dispersion region remains relatively small in the space of parameter values used in previous sections, $(\gamma_{\text{Att}}, \gamma_{\text{Ali}}) \in [0, 0.14] \times [0, 0.4]$. For each strategy, the transition between the dispersion region and the SWARMING phase is located at more or less the same place, $\gamma_{\text{Att}} \approx 0.04$ when $k = 1$ and $\gamma_{\text{Att}} \approx 0.02$ when $k = 2$, and, as in previous section, is where the schooling and milling patterns emerge.

When $k = 1$, the SCHOOLING phase is quite large for $N = 25$, meaning that the burst-and-coast swimming mode is quite able to preserve the schooling formation in such relatively small

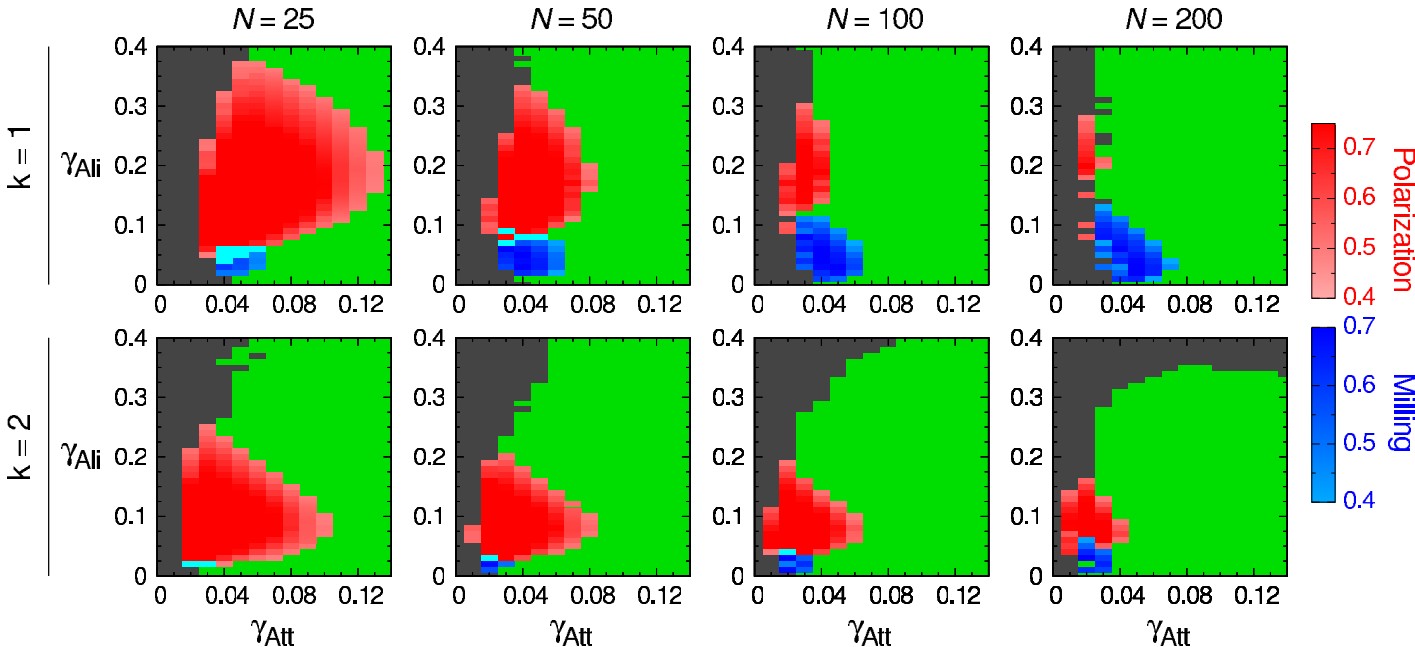

**Fig 8. Phase diagrams for $k = 1$ and $k = 2$ and different group sizes.** SCHOOLING ($P > 0.4$, in red), MILLING ($M > 0.4$, in blue) and SWARMING (green) phases. Cyan regions correspond to $P > 0.4$ and $M > 0.4$, and gray region to high dispersion ($D > 7$ m). Group sizes are, from left to right, $N = 25$, 50, 100, and 200 fish. First row: $k = 1$, second row: $k = 2$. We used $l_{Att} = l_{Ali} = 0.28$ m and $\gamma_R = 0.2$, as for Figs 3 to 6.

groups. Increasing group size by a factor 2 reduces the width of the schooling phase by approximately the same factor: 0.1, 0.06, 0.02, and 0.01 for the successive panels in the first row of Fig 8. The height of the schooling region is also reduced but only moderately (0.32, 0.26, 0.2 and 0.2, although a finer discretization would have been necessary here). This means that preserving a schooling formation is more difficult in large groups when individuals use such a discrete and asynchronous swimming mode.

Most of the region of the parameters that leads to the schooling state in small groups size shrinks and is replaced by the swarming state as $N$ become larger. In parallel, the region leading to a milling state grows. This feature is however in line with the fact that the milling pattern requires minimal group size to emerge [62].

The same observation holds when $k = 2$. However, as group size increases, the shrinking of the region of schooling state and the appearance of milling state in the lower part of that region are not as pronounced as when $k = 1$. Passing from $k = 1$ to $k = 2$ also strongly reduces the range of values of the alignment strength that give rise to ordered phases (*e.g.*, from 0.36 to 0.22 for $N = 25$ and from 0.34 to 0.18 for $N = 50$). In turn, the range of values of the attraction strength $\gamma_{Att}$ for which schooling patterns emerge becomes smaller in small groups and increases with group size: for $N = 25$, the width of the schooling region is smaller when $k = 2$ than when $k = 1$, it is the same for both $k = 1$ and $k = 2$ when $N = 50$, and the relation is reversed in larger groups $N = 100$ and 200, due to the fact that the reduction of the width of the schooling region as $N$ grows is smaller when $k = 2$. Finally, the transition from swarming to dispersion when $\gamma_{Ali}$ is large (above 0.3) is more abrupt in larger groups; see the continuous dispersion map in S2 Fig, where the smooth transition shown for $N = 100$ (slowly decaying as $\gamma_{Ali}$ grows from 0.3 to 0.4) becomes sharp for $N = 200$.

## 4 Discussion

Social interactions play a key role in the ability of groups of individuals to achieve coordinated movements [1, 2, 11, 13]. Identifying these interactions, measuring their effects on individual behavior, and determining the conditions under which they can lead to coordination in a group of individuals are crucial steps to understand collective motion in social groups [63]. Social interactions have been shown to be the result of genetic, epigenetic, endocrine, and neural mechanisms [64–66]. The effect of social interactions on individual behavior is also modulated by different environmental parameters, such as light intensity or water turbidity, that directly affect the range of perception of individuals [46, 67–69]. Finally, the information processing abilities of individuals also deeply impact the effects of these social interactions on collective behaviors that will emerge at the group level [54, 55, 57, 58].

Here we use a data-driven fish school model to quantitatively investigate the impact of perceptual and cognitive factors at the individual level on coordination and the patterns of collective movements. This model that describes the interactions involved in the coordination of burst-and-coast swimming in groups of *H. rhodostomus* has the double advantage of being based on the quantitative characterization of social interactions between fish and their modulation according to the distance, the viewing angle, and the relative orientation between individuals, and of having been validated by experiments with small group sizes [44, 60].

Our results provide a comprehensive understanding of the conditions under which specific collective phases emerge as a function of (1) the way individual fish integrate and respond to the social information conveyed by their conspecifics (*i.e.*, the intensity and spatial range of interactions and the number of influential neighbors), (2) the size of the group, and finally (3) the intensity of random individual behavioral fluctuations. Note that for a given species, these corresponding effective interaction parameters could depend on the age of the fish, their activity, and their environment (water turbidity, light intensity, and hence, time of the day...), which, in addition to the clear theoretical interest, motivated us to study the general phase diagram of our model.

We first find that fish schools remain highly cohesive over a broad range of parameters' values, notwithstanding the discrete and asynchronous burst-and-coast swimming mode. However, whatever the interaction range, the number of influential neighbors selected by fish, and the group size, there must exist a minimal attraction between fish to maintain group cohesion, and a minimal level of alignment is also required to allow the formation of collective patterns such as schooling and milling.

Surprisingly, stronger social interactions do not always enhance group cohesion and coordination: fish school can be less cohesive for high values of attraction and alignment interactions. This phenomenon is quite different from the typical dynamics observed in models with continuous updating rules, in which cohesion and polarization both increase with the intensity of social interactions [31, 62]. Indeed, when a fish chooses its new heading direction at the onset of a kick, high values of $\gamma_{\text{Att}}$ and $\gamma_{\text{Ali}}$ can induce large turning angles. Now, when the angle of deviation is of order $\pi$, this is in fact equivalent to a situation in which a fish chooses its new heading angle randomly. Thus, when the intensity of social interactions becomes too strong, fish move like solitary random walkers, producing a "desocializing" effect that leads to fish dispersion.

This effect is amplified by the number of neighbors with which individuals interact. Considering more influential neighbors amounts to add more social contributions to the instantaneous angle change $\delta\phi$ in Eq (5); if these contributions are high and have the same sign, the final value of $|\delta\phi|$ can more easily be larger than $\pi$. The same happens when the intensity of the individual behavioral fluctuations is high. In that case, a large value of $\gamma_R$ is added to the

contribution to the instantaneous angle change $\delta\phi$ in Eq (4), which leads $|\delta\phi|$ to be larger than $\pi$, thus resulting in the dispersion of fish. Considering the selection pressure in natural conditions, especially under the predator threat, it is likely that the intensity of social interactions has been tuned so that a group can end up in the cohesive region of the phase diagram where its benefits from better adaptiveness by being able to adopt various collective states [70–72].

Social interactions between fish contribute to the spatial organization of the school [73]. By positioning themselves and tuning their headings with respect to their neighbors with independent interactions, individual fish coordinate their collective movements. In the phase diagram, our simulation results have shown that these collective behaviors are located at a thin boundary between a region where the fish disperse and a region of cohesion where the fish are mostly in a swarming state, so that milling and schooling states appear as transition states from dispersed to cohesive groups. This means that small variations in the strength of attraction and alignment allow a school to quickly change from one collective state to another as a response to sudden environmental changes. This is particularly visible at the boundary between the schooling and milling. In that region of the parameter space, the school alternates periods of time during which it is in a schooling state with other periods in which it is in a milling state, an intermittent behavior that was also observed in a data-driven fish school model of *Kuhlia mugil* [62]. These highly coordinated collective states depend on the intensities of social interaction, which could possibly be the result of adaptiveness to environmental stimuli. Indeed, it has been shown that the presence of predators or alarm substances shape the alignment and attraction response of individuals [70, 71, 74, 75], and increase the cohesion of fish schools [73, 76–78]. Moreover, these adaptive changes of social response are heritable and can be reinforced in the offspring generation [72], thus enhancing the fitness of individuals in this species.

Our results also show that the main impact of interacting with the two most influential neighbors is that the ordered phases appear for smaller strengths of attraction and alignment. Moreover, when $k = 2$, the range of values of attraction intensity for which fish adopt a schooling phase is at least twice larger than the one found for $k = 1$, at the same time reducing the milling phase. Indeed, when $k = 2$, the two most influential neighbors are on average located on the front of the focal fish; see S3 Fig. This means that the alignment forces exerted on fish by those neighbors are very weak, so that the contribution of each neighbor to the value of $\delta\phi$ in Eq (5) is small and lead to small heading variation. This particular position of the most influential neighbors facilitates the emergence and the stability of a schooling phase instead of a milling phase, for which a preferred side of attraction is necessary to preserve the clockwise (or counterclockwise) direction of rotation of the school. We have checked this hypothesis at the micro-scale, finding that there exists a strong difference between schooling or a milling state when one consider the position of the most influential neighbors of fish (see S3 Fig). In the schooling state, both influential neighbors are located at the front of the focal fish, while in the milling state, they are mostly located on the sides of fish. Thus, the relative weight of the attraction and alignment interactions leads to different spatial distributions of the most influential neighbors, which in turn conditions the collective phase which emerge at the scale of the school.

Finally, the maximum value of alignment intensity over which schooling appears when $k = 2$ is 2/3 the value found when $k = 1$, meaning that when fish interact with their two most influential neighbors, strong alignment leads to a disorganization of the schooling phase. This is a consequence of the same mechanism described above in which $|\delta\phi|$ reaches high values and fish move like random walkers.

Our simulation results also reveal that individual behavioral fluctuations have two different effects on the size and location of collective states in the phase diagram. On one hand, moderate fluctuations can contribute to trigger spontaneous transitions between schooling and milling states. For instance, the transition from schooling to milling can be triggered by the abrupt

turning of some individuals located in the front of the group (see S5 Video). Then, this change in the moving direction propagates to other individuals through social interactions, so that the whole group starts to rotate with a large radius of curvature. Then, when the individuals who are in the front follow those who are in the back, the group forms a milling phase. Similar mechanisms also happen in the milling-schooling transition, where the fish that trigger the transition can be positioned on the group border (see S6 Video). These examples show that spontaneous transition between schooling and milling can arise from random fluctuations in the moving direction of some few individuals in the school. On the other hand, if the heading fluctuations are too large, the ordered phases of schooling and milling can no longer be maintained and the school switches to a swarming phase. For even larger values of $\delta\phi_{R,i}$, $|\delta\phi|$ in Eq (4) can often become of order $\pi$, thus leading to group dispersion.

Moreover, for a given intensity of the individual behavioral fluctuations, our simulation results indicate that increasing the interaction range ($l_{Att}$ and $l_{Ali}$) has a similar impact on collective dynamics as increasing the strengths of attraction and alignment. Thus, when the interaction range increases, the school remains cohesive and the schooling and milling states appear for smaller values of $\gamma_{Att}$ and $\gamma_{Ali}$. The dispersion due to high values of heading change also occurs for smaller values of the strength of attraction and alignment. As a consequence, the region of ordered states shrinks and shifts to the bottom-left of the parameter space. Transitions between swarming, schooling and milling can be induced by varying the interaction ranges. This is indeed one of the ways used in the most common fish school models that implement repulsion, alignment and attraction rules through layered zones around each fish to get different patterns of collective movements [16, 18, 20, 21]. Furthermore, interaction range is tightly related to the sensory ability of fish. The visual system is the primary sensory modality of fish [79, 80], higher visual acuity and wider visual coverage facilitate group cohesion and polarity [66]. Specifically for *H. rhodostomus*, visual information is essential to enable attraction and alignment with neighbors and thus forming and maintaining schooling state [49]. Accordingly, collective states determined by different interaction ranges in the model correspond to behavioral changes observed when the environment affects the sensory abilities of the fish. This is related to the lower ability of fish to coordinate their behavior in turbid water, which has been observed in many species [46–48].

We also find important differences in collective states caused by variations in group size. We consider size variations in groups from 2 to 8 times larger. Each time the group size $N$ is doubled, the schooling region shrinks by a similar factor. For high values of attraction, schooling phase is replaced by the swarming phase, while for lower values of alignment, the schooling phase is replaced by the milling phase. This effect is clearly visible when $k = 1$, but also happens when $k = 2$. High polarization requires that most of the $N$ fish have the same orientation, *i.e.*, the same heading angle. Heading variations from one fish to a close neighbor are typically small, especially when fish are in a schooling state. If the number of fish is small, heading changes of moderate amplitude can give rise to high polarization. However, in large groups, heading variations between close fish can still be small, but heading differences between distant fish can be very high. Such a disparity of orientations leads to low polarization and therefore to the shrinking of the schooling region in the parameter space. This effect increases with group size. More surprising is the extension of the milling region as group size increases. This result is however in agreement with observations carried out on large fish schools: golden shiners spend significantly more time in a milling state when they are in large groups than when they are in small groups [32]. Our model also shows that *H. rhodostomus* display a schooling state in rather small group sizes (25 individuals for $k = 1$) and a wide range of attraction and alignment strength, suggesting that burst-and-coast swimming mode provides some advantage for coordination and control in small group sizes.

The results obtained here for *H. rhodostomus* can be generalized to other species that perform the same burst-and-coast swimming mode. As recently shown in [45], zebrafish (*Danio rerio*) can be described by a model with an identical structure as the *H. rhodostomus* model, including the way of combining social interactions. The difference between both models, and hence between both species, lies in the actual quantitative shape of the social interaction functions. In particular, the interaction ranges in *H. rhodostomus* are much larger than in *D. rerio*, which can be related to the more social general behavior presented by *H. rhodostomus*. These results show the importance of understanding and characterizing the social interactions in different species that are involved in swimming coordination to better understand the similarities and differences between these coordination mechanisms and the potential impact of ecological constraints on them.

In summary, we have shown that non-trivial collective states can be obtained in a realistic—discrete and asynchronous—model for fish displaying a burst-and-coast swimming mode, and only involving the interaction with 1 or 2 most influential neighbors. Social interactions mediated and constrained by individuals' sensory and cognitive abilities deeply influence collective behaviors that emerge at the group level. By identifying highly synchronized and stable collective states in the parameter space, our results suggest that social responses can be dynamically adjusted in various environmental situations, potentially conferring the social group higher fitness and adaptiveness.

## Supporting information

**S1 Video. Identification of the most influential neighbor in a simulation of a school of fish interacting with their most influential neighbor ($k = 1$).** Individual fish are represented by solid black triangles and vertices indicate the current moving directions. The focal fish is represented in red and its most influential neighbor is represented in dark blue. The video has been slowed down by a factor of 10.
(MP4)

**S2 Video. Identification of the most influential neighbor in a simulation of a school of fish interacting with their two most influential neighbors ($k = 2$).** Individual fish are represented by solid black triangles and vertices indicate the current moving directions. The focal fish is represented in red, its most influential neighbor is represented in dark blue, and its second most influential neighbor is represented in light blue. The video has been slowed down by a factor of 10.
(MP4)

**S3 Video. Schooling state in the fish school model.** Representative example of schooling state corresponding to Fig 4A(I) in a simulation of a school of a hundred fish interacting with their most influential neighbor ($k = 1$). Individual fish are represented by solid black triangles and vertices indicate the current moving directions. Trajectories show the successive positions of individuals over the past 1 s.
(MP4)

**S4 Video. Milling state in the fish school model.** Representative example of milling state corresponding to Fig 4A(II) in a simulation of a school of a hundred fish interacting with their most influential neighbor ($k = 1$). Individual fish are represented by solid black triangles and vertices indicate the current moving directions. Trajectories show the successive positions of individuals over the past 1 s.
(MP4)

**S5 Video. Transition from schooling state to milling state in the fish school model.** The representative transition corresponds to Fig 4A(I) and 4A(II) in a simulation of a school of a hundred fish interacting with their most influential neighbor ($k = 1$). Individual fish are represented by solid black triangles and vertices indicate the current moving directions. Trajectories show the successive positions of individuals over the past 1 s.
(MP4)

**S6 Video. Transition from milling state to schooling state in the fish school model.** The representative transition corresponds to Fig 4A(I) and 4A(II) in a simulation of a school of a hundred fish interacting with their most influential neighbor ($k = 1$). Individual fish are represented by solid black triangles and vertices indicate the current moving directions. Trajectories show the successive positions of individuals over the past 1 s.
(MP4)

**S7 Video. Swarming state in the fish school model.** The representative transition corresponds to Fig 4A(III) in a simulation of a school of a hundred fish interacting with their most influential neighbor ($k = 1$). Individual fish are represented by solid black triangles and vertices indicate the current moving directions. Trajectories show the successive positions of individuals over the past 1 s.
(MP4)

**S1 Fig. Dispersion maps for $k = 1$ and 2 for different combinations of the interaction ranges $l_{Att} = l_{Ali}$ and the intensity of random fluctuations $\gamma_R$.** Each row corresponds to a given value of the interaction range; from top to bottom: $l_{Att} = l_{Ali} = 0.16$, 0.2, 0.24, 0.28, and 0.32 m. Each column corresponds to a given value of random fluctuations; from left to right: $\gamma_R = 0.2$, 0.3, and 0.45. Gray color corresponds to high dispersion ($D > 1$ m), green color to swarming phase. Intermediate color from gray to green correspond to the transition layer from dispersion to swarming. Values extracted from experiments with pairs of *H. rhodostomus* swimming in a small circular tank are $l_{Att} = l_{Ali} = 0.2$ m and $\gamma_R = 0.45$, denoted by a yellow dot in second row, columns 1 ($k = 1$) and 4 ($k = 2$). Figures appear pixelized because simulations are extremely long. We used a fine discretization with $\Delta\gamma = 10^{-2}$ in both axes.
(TIF)

**S2 Fig. Dispersion maps for different group sizes for $k = 1$ and $k = 2$.** Gray color corresponds to region of high dispersion ($D > 0.7$ m), green color to swarming phase. Intermediate color from gray to green correspond to the transition layer from dispersion to swarming. Group sizes are, from left to right, $N = 25$, 50, 100, and 200 fish. First row: $k = 1$, second row: $k = 2$. We used $l_{Att} = l_{Ali} = 0.28$ m and $\gamma_R = 0.2$, as for Figs 3 to 6.
(TIF)

**S3 Fig. Density maps of the relative position of the most influential neighbors of fish when they perform a kick.** (ABC) Density maps of the relative position of the most influential neighbor of a fish when $k = 1$ in (A) the schooling state ($\gamma_{Att} = 0.03$, $\gamma_{Ali} = 0.2$), (B) the milling state ($\gamma_{Att} = 0.04$, $\gamma_{Ali} = 0.04$), and (C) the swarming state ($\gamma_{Att} = 0.1$, $\gamma_{Ali} = 0.05$). (D–I) Density maps of the relative position of (DEF) the most influential neighbor and (GHI) the second most influential neighbor of a fish, when $k = 2$, in (DG) the schooling state ($\gamma_{Att} = 0.02$, $\gamma_{Ali} = 0.1$), (EH) the milling state ($\gamma_{Att} = 0.02$, $\gamma_{Ali} = 0.025$), and (FI) the swarming state ($\gamma_{Att} = 0.1$, $\gamma_{Ali} = 0.05$). Each subgraph corresponds to the average of 2 runs, each run lasting about 1000 s (corresponding to 2000 kicks per fish), in which the first 200 s have been discarded. Bin size is $0.01 \times 0.01$ m$^2$. The focal fish is represented by a short red line located at the

origin (0, 0) and whose heading is pointing to the north. For all cases, $N = 100$, $l_{Att} = l_{Ali} = 0.28$ m, and $\gamma_R = 0.2$.
(TIF)

**S4 Fig. Time evolution of the mean polarization along extremely long runs for a growing attraction strength $\gamma_{Att}$ across the schooling phase.** Each line is the mean polarization of the whole groups of $N = 100$ fish, averaged over 100 runs of 20.000 s of duration, for different values of $\gamma_{Att} \in [0.026, 0.06]$, *i.e.*, across the schooling region shown in Fig 4, for $\gamma_{Ali} = 0.2$. Line color denotes the phase region in which the point ($\gamma_{Att}$, $\gamma_{Ali}$) is located. Time instants are plotted only each 100 times. The value $\gamma_{Att} = 0.034$ seems to be the threshold of the attraction strength below which the curve of the polarization decreases more or less rapidly to the value of no polarization ($\approx 0.1$), and above which the curve seems to remain constant in time, at a smaller height at larger values of $\gamma_{Att}$.
(TIF)

**S5 Fig. Time evolution of the mean number of 3-groups along extremely long runs for a growing attraction strength $\gamma_{Att}$ across the schooling phase.** Each line is the mean number of 3-groups averaged over 100 runs of 20.000 s of duration, for different values of $\gamma_{Att} \in [0.026, 0.06]$, *i.e.*, across the schooling region shown in Fig 4, for $\gamma_{Ali} = 0.2$. Line color denotes the phase region in which the point ($\gamma_{Att}$, $\gamma_{Ali}$) is located. The partition of the $N$ fish into 3-groups is done recursively: start with a fish $i$ and put in its group its three nearest neighbors, then the three nearest neighbors of each nearest neighbor of $i$, and so on. When all the new neighbors of an iteration are already in the group, this 3-group is completed. Take another fish $j$ which is not in a previous 3-group and repeat the process. Once all individuals belong to a 3-group, stop the process and count the number of 3-groups.
(TIF)

**S6 Fig. Probability distribution functions (PDF; solid lines) and corresponding mean values (dashed vertical lines) of the number of different most influential neighbors at a given time in the three collective states, for the case $k = 1$.** (A) Schooling state: $\gamma_{Att} = 0.035$, $\gamma_{Ali} = 0.2$; (B) Milling state: $\gamma_{Att} = 0.04$, $\gamma_{Ali} = 0.05$; (C) Swarming state: $\gamma_{Att} = 0.07$, $\gamma_{Ali} = 0.1$. PDFs and mean values are derived from 30 simulation runs, each of 1000 s duration, and sampled at each second in the time interval [200, 1000] (30 800 = 24000 data sampled per graph, leading to an uncertainty for the corresponding mean of order 0.1%). For all cases, the group size is $N = 100$, the intensity of random fluctuation is $\gamma_R = 0.2$, and the social interaction ranges are $l_{Ali} = l_{Ali} = 0.28$ m. We note that there are significantly more different most influential neighbors in the milling state (mean close to 46) than in the 2 other states (mean around 39).
(TIF)

**S7 Fig. Probability distribution functions (PDF; solid lines) and corresponding mean values (dashed vertical lines) of the number of different nearest neighbors at a given time in the three collective states, for the case $k = 1$.** (A) Schooling state: $\gamma_{Att} = 0.035$, $\gamma_{Ali} = 0.2$; (B) Milling state: $\gamma_{Att} = 0.04$, $\gamma_{Ali} = 0.05$; (C) Swarming state: $\gamma_{Att} = 0.07$, $\gamma_{Ali} = 0.1$. PDFs and mean values are derived from 30 simulation runs, each of 1000 s duration, and sampled at each second in the time interval [200, 1000] (24000 data sampled per graph; uncertainty for the corresponding mean less than 0.1%). For all cases, the group size is $N = 100$, the intensity of random fluctuation is $\gamma_R = 0.2$, and the social interaction ranges are $l_{Ali} = l_{Ali} = 0.28$ m. We find that the mean number of different nearest neighbors is close to 71 in the three collective states, and is hence much larger than the corresponding number of different most influential neighbors.
(TIF)

## Author Contributions

**Conceptualization:** Guy Theraulaz.

**Data curation:** Weijia Wang, Ramón Escobedo.

**Formal analysis:** Weijia Wang, Ramón Escobedo, Clément Sire.

**Funding acquisition:** Zhangang Han, Guy Theraulaz.

**Investigation:** Weijia Wang, Ramón Escobedo, Guy Theraulaz.

**Methodology:** Ramón Escobedo, Clément Sire, Guy Theraulaz.

**Project administration:** Stéphane Sanchez, Zhangang Han, Guy Theraulaz.

**Software:** Weijia Wang, Ramón Escobedo, Stéphane Sanchez.

**Supervision:** Zhangang Han, Guy Theraulaz.

**Validation:** Ramón Escobedo, Clément Sire, Guy Theraulaz.

**Visualization:** Weijia Wang, Ramón Escobedo, Stéphane Sanchez.

**Writing – original draft:** Weijia Wang, Ramón Escobedo, Guy Theraulaz.

**Writing – review & editing:** Weijia Wang, Ramón Escobedo, Stéphane Sanchez, Clément Sire, Zhangang Han, Guy Theraulaz.

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
