## [Decision Letter · Decision Letter 0]

21 Nov 2021

Dear Dr. Theraulaz,

Thank you very much for submitting your manuscript "The impact of individual perceptual and cognitive factors on collective states in a data-driven fish school model" for consideration at PLOS Computational Biology.

As with all papers reviewed by the journal, your manuscript was reviewed by members of the editorial board and by several independent reviewers. In light of the reviews (below this email), we would like to invite the resubmission of a significantly-revised version that takes into account the reviewers' comments.

We cannot make any decision about publication until we have seen the revised manuscript and your response to the reviewers' comments. Your revised manuscript is also likely to be sent to reviewers for further evaluation.

Sincerely,

Nir Gov

Associate Editor

PLOS Computational Biology

Natalia Komarova

Deputy Editor

PLOS Computational Biology

Reviewer's Responses to Questions

**Comments to the Authors:**

Reviewer #1: Review is uploaded as attachment.

Reviewer #2: The authors study numerically the properties of a flocking model that they used previously to describe fish school data. The focus is on the impact of modeling elements that the authors interpret as regulating perceptual and cognitive factors. Special attention is paid to the number of number of influential neighbors, limiting this number to even two and one. Collective behavior is characterized using two observables, the polarization and the milling index, in small group size with up 200 individuals (small in comparison to studies that aimed at characterizing the large scale properties, and “universal classes”, of flocking models).

I found the results of interest, providing insight on the collective dynamics of fish schools of medium, and the manuscript clearly written.

There is, however, one issue that the authors could address, which is at the moment missing in the manuscript.

The model has attraction and repulsion and both interactions are formulated via complicated expressions that depend on relative distance, orientation, and position, involving a large number of parameters (even if here several of them are kept constant). Furthermore, the authors indicate in the conclusions “Transition between swarming, schooling, and milling can be induced by varying the interaction ranges”. While I do not disagree with this observation, I think it would be important to mention that it has been shown that these collective behaviors can be also induced in simple models that use a fixed interaction range and only attraction (i.e. no alignment) by varying the angle of the field of view, see Barberis et al. Phys. Rev. Lett. 117, 248001 (2016), where the transitions to different collective behaviors can be understood as instabilities. Moreover, I wonder whether it is really necessary to have both, attraction and repulsion, and complicated expressions for both. I am not asking the authors to answer to this question (which I image require a separate study), but to comment on potential alternatives to obtain different collective dynamics, which could be of relevance in the study of animal groups (including fish groups).

Reviewer #3: This paper describes models for fish schooling that are inspired by a data-drive model based on the collective behaviour of H. rodostomus. This work goes to great detail in describing how the different model parameters influence the formation of different collective states. Further, these results are not just presented as is but, rather, each one is accompanied by the relevant intuitive explanation. The discussion nicely wraps up the paper by connecting these findings to relevant biological questions concerning fish schooling. In general, I would support publication after some of my concerns, listed below, have been lifted.

1. It is not clear enough if variation in different model parameters is meant to describe different fish species or dynamic variations in the species of study itself.

For the first case (describe a variety of species), how generalizable is this model expected to be for other species? The model is quite unique in that that the most influential fish are not the nearest ones (either metrically or topologically) but rather those that are most different from the focal fish. It might be the case that other fish species use very different modes of schooling which would make much of the results here insignificant. Do the authors expect this behavior to appear in other species? What is the evidence for this?

In the second case (i.e the model is meant to describe variations in the same species), is there any evidence that these fish vary their interaction parameters in different circumstanes? Which parameters are more likely to be controlled by the fish and which by the environment? Are all parameters studied in this work of equal importance? If these parameter changes are not experimentally approachable then why should they interest the reader?

2. The paper is tedious to read. More and more parameters, more and more phase diagrams, and more and more intuitions and discussions of this or that effect whose relative importance (by whichever scale) is not really mentioned. I think that this way of presenting the results does not do the work justice. One possible solution is to organize the results and parameters by their effects on relevant behaviors such as the ability to escape from a predator as discussed in the introduction.

3. It would be nice to mark the region that corresponds to the fitted parameters relevant to H. rhodostomus on all or some of the phase space plots where.

4. Specific events that trigger a change in the collective state are shown in the the SI movies and shortly mentioned in the discussion but not quantitative analysis of which events trigger a change and which don’t – how stochastic this is etc…. this is provided in the results. Reuslts of this sort were previously presented by the Couzin group (https://doi.org/10.1371/journal.pcbi.1002915) and by the Gov group (https://doi.org/10.1371/journal.pcbi.1006068). In the discussions the authors suggest that H. rodostomus may change its collective state by varying some internal parameters – is there any evidence for this? Again, this kind of control can make the paper a more engaging read.

5. The discussion attributes some of the findings in this paper to the fact that the fish exhibit discrete rather than continuous update (line 574). However, I did not find substantiation for this in the results. Rather, it could be that other characteristics of the model (such as the identity of the most influential fish as the most different one) account for the results. In short, this claim is very interesting and must be substantiated.

6. There are several small prediction verifications mostly hidden in the SI and mostly mentioned in the discussion. I think that analyzing a model to this degree of detail but not testing a single one of its predictions makes the paper somewhat lopsided and not as strong as it could have been. Predictions concerning the species of study but varying group size or water turbidity (as mentioned in the discussion) are certainly possible. The journal policies state that “reliability and significance of biological discovery through computation should be validated and enriched by experimental studies. Inclusion of experimental validation is not required for publication, but should be referenced where possible” So, I would simply urge the authors to highlight any experimental tests/validations to any non-trivial predictions of their model.

**Have the authors made all data and (if applicable) computational code underlying the findings in their manuscript fully available?**

Reviewer #1: None

Reviewer #2: Yes

Reviewer #3: Yes

PLOS authors have the option to publish the peer review history of their article (what does this mean?). If published, this will include your full peer review and any attached files.

Reviewer #1: No

Reviewer #2: No

Reviewer #3: No
---

## [Decision Letter · Decision Letter 1]

2 Feb 2022

Dear Dr. Theraulaz,

Thank you very much for submitting your manuscript "The impact of individual perceptual and cognitive factors on collective states in a data-driven fish school model" for consideration at PLOS Computational Biology. As with all papers reviewed by the journal, your manuscript was reviewed by members of the editorial board and by several independent reviewers. The reviewers appreciated the attention to an important topic. Based on the reviews, we are likely to accept this manuscript for publication, providing that you modify the manuscript according to the review recommendations.

Sincerely,

Nir Gov

Associate Editor

PLOS Computational Biology

Natalia Komarova

Deputy Editor

PLOS Computational Biology

[LINK]

Reviewer's Responses to Questions

**Comments to the Authors:**

Reviewer #1: I have read the revised manuscript, as well as the responses to all my points/concerns. I thank the authors for the very detailed responses, particularly the response to my 3rd point. I recognize and appreciate the effort made on computing the probability distributions for the different most influential neighbors (DMIN) and the different first nearest neighbors (DFNN), which indeed responds to what I had in mind when raising my question.

Therefore, I think the manuscript is ready to be published in PLOS Computational Biology.

Reviewer #2: I wrote an initial friendly report, trying to be as positive as possible. Despite my comments could have been presented as a serious criticism, I chose in my report to be constructive and nice to the authors. Unfortunately, it seems that the authors decided not to take seriously my comments, providing a vague answer and avoiding adding comments (even a couple of short sentences) to address them. I find difficult to understand the authors’ attitude.

1. My comment on the group sizes used in the manuscript was not at all a criticism. If the authors looked at larger groups in another publication is irrelevant for the current manuscript, except for placing the current study in context.

2. In quantum mechanics or animal behavior, “successful” mathematical models are supposed to reproduce (or describe) empirical data and should allow to formulate predictions that can be used to test the validity of the model: the scientific method allows us to only falsify a theory, but not to validate it. So, at a given time, we have at most a best candidate model. My question was about the uniqueness of the model. In my opinion, it is not evident that the data cannot be described using a reduced set of interactions and/or with a different type of formulation. I just suggested the authors to comment on this point.

3. In their answer the authors comment that the model is based on one of their previous publications and that “social interactions functions of fish have been specifically extracted from experimental data [...] don’t leave room for manipulation without the risk of losing fidelity of what is observed in the reality”. What are the author pretending here? Are they suggesting that their model is independent of the choice of observables, dynamic variables, mathematical tools, or types of equation (e.g. standard ODEs, etc)? Are they suggesting that there exist a unique description of “the reality”? Or do they claim that their model is “the reality”? And by the way, what is “the reality”?

The level of detail to be incorporated in a model is also a choice. For instance, the burst-and-coast swimming mode can be relevant depending on the question/observable to be tackled, as much as is friction is relevant to understand how a pendulum stops swinging, but we learn a great deal by neglecting it and focusing on short time scales (what is the reality here?). I believe that it is evident that idealized models have proven to be very useful. So, what is the question that the authors try to address that requires the explicit incorporation of the burst-and-coast swimming mode. And is it correct to think that models with explicit burst-and-coast are so fundamentally different from other models when it comes to group polarization and/or cohesion that it is not possible to compare them with previously existing models with no burst-and-coast?

Reviewer #3: The authors have adequately addressed my remark.

I still hold it that, organized as is, the paper is a tedious read and that, unfortunately, this would reduce its impact but as this is a stylistic comment comment i just leave it with the authors.

**Have the authors made all data and (if applicable) computational code underlying the findings in their manuscript fully available?**

Reviewer #1: Yes

Reviewer #2: Yes

Reviewer #3: Yes

PLOS authors have the option to publish the peer review history of their article (what does this mean?). If published, this will include your full peer review and any attached files.

Reviewer #1: No

Reviewer #2: No

Reviewer #3: No

Figure Files:

Data Requirements:

Reproducibility:

References:

---

## [Editor Report · Decision Letter 2]

8 Feb 2022

Dear Dr. Theraulaz,

We are pleased to inform you that your manuscript 'The impact of individual perceptual and cognitive factors on collective states in a data-driven fish school model' has been provisionally accepted for publication in PLOS Computational Biology.

Best regards,

Nir Gov

Associate Editor

PLOS Computational Biology

Natalia Komarova

Deputy Editor

PLOS Computational Biology

---

## [Editor Report · Acceptance letter]

24 Feb 2022

PCOMPBIOL-D-21-01647R2 

The impact of individual perceptual and cognitive factors on collective states in a data-driven fish school model

Dear Dr Theraulaz,

I am pleased to inform you that your manuscript has been formally accepted for publication in PLOS Computational Biology. Your manuscript is now with our production department and you will be notified of the publication date in due course.

With kind regards,

Katalin Szabo
